# *NextQuill*: Causal Preference Modeling for Enhancing LLM Personalization

Xiaoyan Zhao[1]* Juntao You[2]* Yang Zhang[3]† Wenjie Wang[2]† Hong Cheng[1]
Fuli Feng[2] See-Kiong Ng[3] Tat-Seng Chua[3]
[1]The Chinese University of Hong Kong [2]University of Science and Technology of China
[3]National University of Singapore
xzhao@se.cuhk.edu.hk, ustcyjt@mail.ustc.edu.cn

## Abstract

Personalizing large language models (LLMs) is increasingly important as they are progressively integrated into real-world applications to support users' daily lives. However, existing approaches often fail to distinguish which components of response predictions by model and ground-truth response in training data truly reflect user preferences, resulting in shallow personalization alignment. In this paper, we introduce NextQuill, a novel LLM personalization alignment framework grounded in causal preference modeling. We approach personalization from a causal perspective, recognizing that model-predicted responses (model side) and user-written ground-truth responses (data side) are both outcomes shaped by user history (characteristics) and other context factors. To better capture user preferences, we define causal preference effects as the causal effect of the user history/characteristics on outcomes from the model/data side. Building on this foundation, NextQuill introduces two complementary alignment strategies: (1) aligning model-side causal preference effects (on predictions) with those of ground-truth data, rather than indiscriminately aligning all predictions, and (2) emphasizing learning the preference-driven ground-truth tokens, identified via data-side causal preference effects, rather than treating all tokens equally. As such, NextQuill shifts the alignment process toward learning from causal preference effects, facilitating more effective and personalized LLM adaptation. Experiments on multiple personalization benchmarks demonstrate that NextQuill substantially improves personalization quality. Code is available at https://github.com/juntaoyou/NextQuill.

## 1 Introduction

Large Language Models (LLMs) have exhibited exceptional capabilities across various domains (Achiam et al., 2023; Zhao et al., 2024), driving their widespread deployment in real-world applications (Christakopoulou et al., 2023; Liu et al., 2025b; Shi et al., 2024; Zhao et al., 2026) such as virtual assistants and content generation systems. However, existing LLMs are typically designed under a general-purpose "one-size-fits-all" paradigm (Qiu et al., 2025b), neglecting the diversity of user preferences in aspects such as needs, goals, and communication styles. As LLMs become more integrated into individuals' daily lives and work, accounting for personal preferences in LLMs is becoming increasingly important for delivering tailored and engaging experiences (Mysore et al., 2024; Liu et al., 2025a). This growing demand has sparked a surge of research interest in *LLM personalization*, with notable efforts emerging across both academic research and industry (Alaluf et al., 2024; Pham et al., 2024; Li et al., 2023).

Intuitively, user historical data inherently reflects individual preferences, playing a critical role in enabling personalization for LLMs. Based on how this data is utilized, existing personalization approaches can be broadly classified into two categories. The first follows a memory-retrieval paradigm (Salemi et al., 2024a; Qiu et al., 2025b; Zhuang et al., 2024), in which user history is

---

*Equal contribution.
†Corresponding author. Email: zyang1580@gmail.com, wenjiewang96@gmail.com.

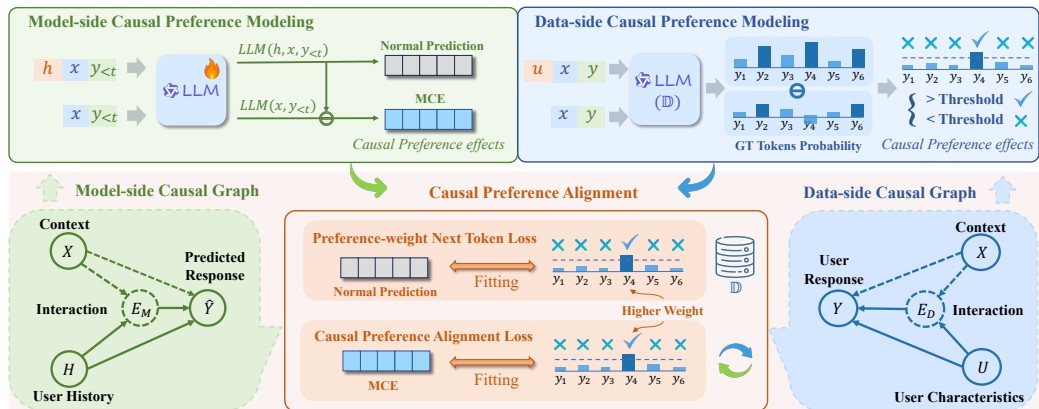

Figure 1: Illustration of NextQuill. The model-side causal graph depicts the response prediction process of LLMs, in which the outcome is influenced by user history, context information, and their interaction. The data-side causal graph describes how users generate the ground-truth response, which is jointly influenced by user characteristics, the context, and their interaction. Based on these, NextQuill introduces the causal preference modeling, defining causal preference effects on both model and data sides, and accordingly introduces two causal preference alignment strategies (*i.e.,* two losses) to enhance the personalization of LLMs.

stored in an external memory and relevant information is then dynamically retrieved and injected into the prompt to guide generation. In this setting, LLMs rely on steering prompts to produce user-aligned responses. However, performance may be limited due to the lack of alignment mechanisms specifically designed for personalization. The second category involves further fine-tuning LLMs with historical data to adapt model parameters (Zhang et al., 2023a; Tan et al., 2024b; Liu et al., 2024b) — typically, tuning the model to leverage past user behaviors for predicting subsequent ones. This fine-tuning process customizes the model's capacity to capture individual preferences, facilitating more explicit personalization alignment.

Although existing tuning-based methods improve alignment, we argue that their alignment remains suboptimal due to superficial modeling and the uncritical use of user data, overlooking what truly matters for preference modeling. From the perspective of preference representation within the model, these methods typically treat all predictions generated from the entire input as inferred preferences and align them uniformly with the ground truth. This overlooks the fact that it is primarily the inferences derived from historical behavioral data that genuinely reflect the model's internal preference modeling. From the data supervision perspective, these methods treat all tokens in the ground-truth responses equally, failing to account for the unequal contribution of different tokens to the expression of user preferences. Such shallow treatment prevents the model from identifying and emphasizing the preference-driven components critical for personalized generation, limiting the alignment quality.

To address the issues, we must determine what truly matters in preference modeling. We approach this from a causal perspective, treating the generation of predicted responses (by model) and the creation of ground-truth responses (by users) as outcomes of an underlying causal process, separately depicted by the left and right causal graphs in Figure 1. As shown, both the generation of model predictions and the creation of ground-truth responses can be influenced by non-preference factors—beyond user history for model predictions and beyond user characteristics for ground-truth responses, mainly contextual factors. However, in theory, only the component caused by preference-related factors can accurately represent the user preference. To isolate the preference-driven components on both the model and data sides, we propose Causal Preference Modeling, grounded in causal effect estimation. Specifically, on the model side, we define a causal preference effect as the causal effect of historical user data on model predictions, representing the true internal preferences encoded by the LLM. On the data side, we define a causal preference effect as the causal effect of user characteristics on the creation of ground-truth responses, measuring the extent to which each token in the ground-truth is driven by user preferences.

Building on this foundation, we propose *NextQuill*, a novel causal preference modeling-based alignment method for LLM personalization. NextQuill introduces two strategies to highlight preference-driven components during alignment on both the model and data sides. On the model side, we use causal preference effects to isolate the preference-driven components of model predictions, representing the preferences internalized by the LLM, and then introduce a new loss function to align these effects with those in the ground truth—rather than indiscriminately aligning all predictions. On the data side, we utilize causal preference effects to define a causal attribution score for the ground-truth data tokens, identifying those that are truly driven by user preferences. We then assign higher attention weights to these tokens during data fitting, ensuring that preference alignment focuses on the most relevant, preference-driven tokens. By integrating these two strategies, NextQuill shifts the alignment process toward learning from causal preference effects, enabling more effective and personalized LLM adaptation.

The main contribution of this work can be summarized as follows:

- We introduce a causal perspective on LLM personalization and propose a causal preference modeling approach, defining causal preference effects at both model and data sides.
- We propose NextQuill, a causal preference modeling-based alignment method that enhances LLM personalization by 1) aligning model-internal causal preference effects with those in ground-truth response, rather than indiscriminately aligning all predictions and 2) focusing learning on preference-driven ground-truth tokens, rather than treating all tokens uniformly.
- We conduct comprehensive experiments across multiple personalization domains, showing that NextQuill significantly improves generation quality and overall personalization performance.

## 2 PROBLEM FORMULATION

We study the task of personalized text generation, which aims to enhance an LLM $\mathcal{M}_\theta$ by incorporating user-specific information to generate responses tailored to individual users. Given a user query $x$, which specifies the item or objective the user focuses on, a general LLM produces a response $\hat{y} = \mathcal{M}_\theta(x)$. To enable personalization, we assume each user is associated with historical text information $h$ that reflects their preferences when having the query. The ground-truth response is denoted by $y$, representing text written by the user. Then, the triplet $(x, h, y)$ forms a data sample for the user, and the collection of such samples constitutes the user dataset $\mathbb{D}$, having $(x, h, y) \in \mathbb{D}$. Our goal is to develop a model that utilizes both the input query $x$ and the associated profile $h$ to generate a personalized response $\hat{y} = \mathcal{M}_\theta(x, h)$. Specifically, we aim to: (1) improve the preference modeling ability of $\mathcal{M}_\theta$ by effectively incorporating personalized information from user history $h$; (2) better align the generated response $\hat{y}$ with the personalized information embedded in the expected response $y$. Ultimately, improve the $\mathcal{M}_\theta$ to generate personalized responses $\hat{y}$.

## 3 METHODOLOGY

To improve LLM personalizaiton, we must determine what truly matters in preference modeling. At the beginning of the section, we conduct a causal analysis to address this question, introducing a new perspective of causal preference modeling. Following this, we present our NextQuill method, which explicitly emphasizes causal preference effects on both the model and data side during training to improve LLM personalization.

### 3.1 CAUSAL PREFERENCE MODELING

The core of causal preference modeling lies in analyzing the processes of response prediction by the model (model-side) and ground-truth response generation by users (data-side) using *causal graph* (Pearl, 2009), deriving the causal effects related to preference factors as the basis for preference modeling. We next provide the analysis from the two sides, respectively.

### 3.1.1 MODEL-SIDE CAUSAL PREFERENCE MODELING

By definition, a *causal graph* is a directed acyclic graph (DAG), where each node represents a variable and each edge indicates a causal relationship between two variables (Zhou et al., 2024). It serves as a

powerful tool for describing the processes of data generation or model prediction, thereby guiding method design (Zhang et al., 2021).

**Model-side Causal Graph.** In our work, we use the left causal graph in Figure 1 to illustrate the response prediction process of LLMs, comprising the following nodes:

- $H$: Historical user data that reflects user preferences.
- $X$: Other context information, *e.g.,* query and task prompt.
- $E_M$: The latent variable that captures the interaction between $X$ and $H$ in the model.
- $\hat{Y}$: The predicted response.

Due to the complex prediction pathway in the model, $H$ and $X$ can each independently influence the predicted response or interact with each other to do so, as represented by the following edges:

- $H \rightarrow \hat{Y}$ & $X \rightarrow \hat{Y}$: $H$ and $X$ can independently affect $\hat{Y}$.
- $(H, X) \rightarrow E_M \rightarrow \hat{Y}$: $H$ and $X$ interact with each other to affect $\hat{Y}$.

As shown, both $H$ and $X$ exert causal effects on $\hat{Y}$. However, it is the effects of $H$—driven by user information—that are most indicative of preference signals. Therefore, we need to identify these causal effects as the key to effective preference modeling, which we term model-side causal preference effects.

**Model-side Causal Preference Effect.** Based on causal theory and the corresponding causal graph, the target causal effect of $H = h$ (conditioned on a given $x$) can be formulated as:

$$
\begin{aligned}
MCE(\hat{Y}_t|h, x) &= P(\hat{Y}_t \mid H = do(h), x) - P(\hat{Y}_t \mid H = do(0), x) \\
&= P(\hat{Y}_t \mid H = h, x) - P(\hat{Y}_t \mid H = 0, x),
\end{aligned}
\tag{1}
$$

where $MCE(\hat{Y}_t|h, x)$ denotes the model-side casual effects, $\hat{Y}_t$ denotes the $t$-th token for the prediction, and $do(\cdot)$ denotes the $do$-calculus. The expression $H = do(h)$ represents an intervention that sets $H$ to the specific value $h$, while $H = do(0)$ denotes setting $H$ to a reference value (*e.g.*, a null). The term $P(\hat{Y}_t \mid H = do(h), x)$ refers to the interventional probability, whereas $P(\hat{Y}_t \mid H = h, x)$ is the corresponding observational (Bayesian) probability. Under our causal graph, these two are equivalent, *i.e.,* $P(\hat{Y}_t \mid H = do(h), x) = P(\hat{Y}_t \mid H = h, x)$, and similarly for the reference case.

**Interpretation:** Functionally, this model-side causal effect measures the part of the predictions that is truly driven by the preferences represented by the user's history, *reflecting the true internal preferences modeled (or captured) by LLMs* from inputs. This suggests that not all parts of the model's predictions should be treated equally.

### 3.1.2 Data-side Causal Preference Modeling

**Data-side Causal Graph.** The right causal graph in Figure 1 depicts the ground-truth response generation process (*i.e.*, how users produce the ground-truth response). The rationale for this causal graph is explained as follows:

- $U$: User characteristics, which indicate the user's underlying preference characteristics.
- $X$: Context information, mainly the item or the objective the user focuses on.
- $E_D$: The latent variable that captures the interaction between $U$ and $X$.
- $Y$: The outcome variable, representing the user-written ground-truth response.

The causal relationships between these variables are described by the following edges:

- $(U, X) \rightarrow E_D \rightarrow Y$: $U$ and $X$ can influence $Y$ through their interaction $E_D$, indicating that when generating $Y$, the user engages with the context $X$ to focus on the parts of interest.
- $X \rightarrow Y$: $X$ can influence $Y$ independently of $U$; for instance, no matter who the user is, the lead actor is often mentioned in the reviews for a specific movie.

- $U \to Y$: $U$ can influence $Y$ independently of $X$. For example, regardless of the content $X$, a user may consistently use the same catchphrases.

Similarly, both $U$ and $X$ have causal effects on $Y$, but it is the effects of $U$—reflecting user preferences—that most directly convey preference signals. Identifying $U$'s causal influence on $Y$ is therefore essential for isolating preference in the ground-truth response. We refer to these causal effects as the data-side causal preference effects.

**Data-side Causal Preference Effect.** Similar to the model side, the data-side causal effect of $U = u$ on $Y$ (conditioned on a given $x$) can be formulated as:

$$
\begin{aligned}
DCE(Y_t|u,x) &= P(Y_t \mid U = do(u), x) - P(Y_t \mid U = do(0), x) \\
&= P(Y_t \mid U = u, x) - P(Y_t \mid U = 0, x),
\end{aligned}
\tag{2}
$$

where $Y_t$ denotes the $t$-th token of $Y$, $DCE(Y_t \mid u, x)$ denotes the obtained data-side causal effects, and $u$ represents the characteristics of the specific user.

**Analysis**: Given a sample $(x, h, y) \in \mathbb{D}$, by specifying $Y_t$ as the $t$-th token of $y$ (*i.e.*, $y_t$), the causal effect $DCE(Y_t = y_t \mid u, x)$ *quantifies the extent to which the generation of $y_t$ is driven by user preferences*. A higher value indicates that the token more strongly reflects the user's preferences. This implies that different tokens vary in their preference relevance and should be treated differently during training.

## 3.2 Personalization Alignment based on Causal Preference Modeling

Building on our analysis of the causal preference modeling framework, we find that different parts of both model predictions on the response and ground-truth response vary in their relevance to user preferences. To enhance personalization alignment, it is crucial to identify the preference-driven components based on causal preference effects and explicitly emphasize their learning. Specifically, once identified, we adopt two key strategies to strengthen their influence: 1) on the one hand, we prioritize learning from ground-truth tokens that are driven by user preferences; 2) on the other hand, we introduce a causal preference alignment loss to enforce alignment between the true internal preferences modeled by LLMs (model-side causal preference effects) and those in the ground-truth data (data-side causal preference effects). Figure 1 provides an overview of our method. In the following, we first describe how to identify preference-driven components, followed by the tuning strategies.

### 3.2.1 Preference-driven Component Identification

Section 3.1 provides key insights for identifying preference-driven components in both model predictions and ground-truth responses. On the model side, we can use the causal preference effects on predictions to extract the parts truly reflecting user preferences. On the data side, we can assess the causal preference effects for each ground-truth token to determine whether it is preference-driven.

**Identification on Model Side.** On the model side, we leverage the causal preference effect of user history on predictions, as $MCE$ defined in Equation 1, to identify the components of predictions that reflect user preferences. The causal effect in Equation 1 is defined from a probabilistic perspective, and it must be converted into an empirical form for practical application. Given a data sample $(x, h, y) \in \mathbb{D}$, the empirical causal effect for the prediction of the $t$-th token is computed as:

$$
f_\theta(x, h, y_{<t}) - f_\theta(x, \emptyset, y_{<t}),
\tag{3}
$$

where: 1) $f_\theta(x, h, y_{<t})$ denotes the model's prediction for the $t$-th token, conditioned on the user query $x$, user history $h$, and the preceding tokens $y_{<t}$; and 2) $f_\theta(x, \emptyset, y_{<t})$ denotes the prediction when the user history is removed. Besides, $\theta$ denotes the learnable model parameters of LLMs. Notably, this formulation differs slightly from Equation 1 in that we include $y_{<t}$ as part of the input. This is because $y_{<t}$ can be considered part of the evolving context (or extended query) during generation of the $t$-th token. For simplicity, we do not merge $x$ and $y_{<t}$ explicitly in notation.

**Identification on Data Side.** Given a sample $(x, h, y) \in \mathbb{D}$, the causal effect $DCE(Y_t = y_t \mid u, x)$ in Equation 2 measures the extent to which the generation of the $t$-th token in $y$ (*i.e.*, $y_t$) is driven by

user preferences. Based on this, we classify each ground-truth token $y_t$ as either preference-driven or not. To represent this distinction, we assign a token weight $\omega_t$ to denote the results. Specifically,

$$\omega_t = \begin{cases} \lambda, & \text{if } DCE(Y_t = y_t \mid u, x) > \delta \\ \epsilon, & \text{otherwise} \end{cases}, \tag{4}$$

where $\delta$ denotes the threshold. If $\omega_t = \lambda$, the token is a preference-driven token; otherwise, it is treated as a non-preference-driven token. $\lambda$ and $\epsilon$ are two manually defined weights, which will be used later in the tuning process.

*Estimating $DCE(Y_t = y_t \mid u, x)$.* To estimate $DCE(Y_t = y_t \mid u, x)$ for a user with characteristics $u$, we use an LLM that has seen dataset $\mathbb{D}$ for approximation in a way similar to Equation 3. Moreover, since the original user characteristics $u$ are not directly available, we represent them using the historical data $h$. The estimation is then given by:

$$DCE(Y_t = y_t \mid u, x) \approx f_{\theta_{\mathbb{D}}}(y_t|x, h, y_{<t}) - f_{\theta_{\mathbb{D}}}(y_t|x, \emptyset, y_{<t}), \tag{5}$$

where $\theta_{\mathbb{D}}$ denotes the model parameters of the LLM having seen $\mathbb{D}$. Here, $f_{\theta_{\mathbb{D}}}(y_t \mid x, h, y_{<t})$ estimates the probability that the $t$-th token is $y_t$. Since the LLM has seen $\mathbb{D}$, $f_{\theta_{\mathbb{D}}}(y_t \mid x, h, y_{<t})$ can serve as an approximated estimation for $P(Y_t \mid U = h, X)$ in Equation 2. Appendix C.3 discusses two practical implementations, including both dynamic and fixed DCE estimation.

### 3.2.2 TUNING STRATEGY

To enhance learning on preference-driven components in both model predictions and ground-truth data, we adopt two strategies: (1) assign higher weights to preference-driven tokens during ground-truth response fitting, and (2) introduce a new causal preference alignment loss to align model-side causal preference effects and the data-side ones, *i.e.,* align MCE and DCE. The weighting mechanism is applied to both the original standard next token prediction loss and the new causal preference alignment loss. We finally combine the two losses to fine-tune the model like SFT.

**Preference-weighted Next Token Loss.** We adopt the normal next-token prediction loss to ensure coherent and appropriate text generation, while modifying it into a preference-weighted form to better emphasize the learning of preference-driven tokens, guided by the DCE-driven weights $\omega_t$ defined in Equation 4. Specifically, the weighted loss $L_n$ is formulated as:

$$L_n = \frac{1}{|\mathbb{D}|} \sum_{(x,h,y) \in \mathbb{D}} \sum_{t=1}^{|y|} \omega_t \cdot \ell(f_\theta(x, h, y_{<t}), y_t), \tag{6}$$

where $\ell(\cdot)$ is the standard cross-entropy loss, and $|y|$ is the length of $y$.

**Causal Preference Alignment Loss.** To enhance personalization learning, we further align the data-side causal preference effects (DCE) with those reflected in the model (MCE), ensuring that the model's internal preference representation matches the preferences expressed in the ground-truth response. To achieve this, we introduce a new *causal preference alignment loss*. Specifically, we leverage the preference-driven component of the prediction, as defined in Equation 3, to align with the ground-truth data, placing greater emphasis on fitting the preference-driven tokens. The optimization objective $L_p$ is formulated as follows:

$$L_p = \frac{1}{|\mathbb{D}|} \sum_{(x,h,y) \in \mathbb{D}} \sum_{t=1}^{|y|} \omega_t \cdot \ell\big(f_\theta(x, h, y_{<t}) - f_\theta(x, \emptyset, y_{<t}); y_t\big). \tag{7}$$

This loss directly encourages alignment between the LLM's internal causal preference effects and the causal preference effects in the ground-truth.

**Final Optimization Objective.** We combine the two losses to fine-tune the model in a manner similar to SFT. The optimization objective is formulated as:

$$\theta^\star = argmin_\theta \quad L_n + \alpha \cdot L_p, \tag{8}$$

where $\alpha$ is a hyperparameter to balance $L_n$ and $L_p$, and $\theta^\star$ denotes the obtained model parameters.

Table 1: Main results of our NextQuill method on four benchmark datasets. **Bold** numbers denote the best performance, while underlined numbers denote the second best.

| Datasets | Methods (→) | Base | Retrieval-based Methods | | | | | PEFT-based Methods | | | |
|---|---|---|---|---|---|---|---|---|---|---|---|
| | Metrics (↓) | Qwen | Contriever | LatestK | LLM-TRSR | CoS | SFT | OPPU | ContextSFT | NextQuill |
| Book Review | ROUGE-1 | 0.0519 | 0.0507 | 0.0752 | 0.0465 | 0.0455 | 0.0717 | 0.1502 | 0.1661 | **0.2318** |
| | ROUGE-L | 0.0267 | 0.0275 | 0.0406 | 0.0269 | 0.0278 | 0.0333 | 0.0750 | 0.0836 | **0.1270** |
| | METEOR | 0.0639 | 0.0599 | 0.0851 | 0.0470 | 0.0337 | 0.0884 | 0.1013 | 0.2158 | **0.2580** |
| | BLEU | 0.0591 | 0.2064 | 1.3314 | 0.5733 | 0.3095 | 0.1849 | 0.6935 | 2.1214 | **3.5718** |
| | BERTScore | 0.7385 | 0.7414 | 0.7235 | 0.7997 | 0.7141 | 0.7841 | 0.8020 | 0.8013 | **0.8182** |
| Movie Review | ROUGE-1 | 0.0470 | 0.0516 | 0.0527 | 0.0924 | 0.0145 | 0.0599 | 0.1231 | 0.1573 | **0.2015** |
| | ROUGE-L | 0.0255 | 0.0290 | 0.0292 | 0.0481 | 0.0085 | 0.0301 | 0.0648 | 0.0753 | **0.1041** |
| | METEOR | 0.0575 | 0.0580 | 0.0582 | 0.0918 | 0.0107 | 0.0697 | 0.0796 | 0.1718 | **0.1918** |
| | BLEU | 0.0402 | 0.2454 | 0.3178 | 1.2885 | 0.0091 | 0.1422 | 0.2555 | 1.7151 | **2.3845** |
| | BERTScore | 0.7354 | 0.7411 | 0.7395 | 0.7373 | 0.6731 | 0.7445 | 0.7970 | 0.7943 | **0.8064** |
| CD Review | ROUGE-1 | 0.0438 | 0.0501 | 0.0513 | 0.1074 | 0.0327 | 0.0692 | 0.1405 | 0.1505 | **0.1976** |
| | ROUGE-L | 0.0231 | 0.0271 | 0.0282 | 0.0566 | 0.0176 | 0.0324 | 0.0714 | 0.0714 | **0.0998** |
| | METEOR | 0.0517 | 0.0545 | 0.0560 | 0.0976 | 0.0234 | 0.0768 | 0.0848 | 0.1583 | **0.1805** |
| | BLEU | 0.0330 | 0.2386 | 0.2688 | 1.2250 | 0.0780 | 0.1220 | 0.3090 | 1.3487 | **1.9676** |
| | BERTScore | 0.7311 | 0.7380 | 0.7375 | 0.7479 | 0.6900 | 0.7457 | 0.7988 | 0.7901 | **0.8060** |
| Topic Writing | ROUGE-1 | 0.0684 | 0.0536 | 0.0498 | 0.0618 | 0.0523 | 0.0345 | 0.1229 | 0.0934 | **0.1510** |
| | ROUGE-L | 0.0353 | 0.0286 | 0.0263 | 0.0362 | 0.0276 | 0.0198 | 0.0621 | 0.0444 | **0.0729** |
| | METEOR | 0.0520 | 0.0592 | 0.0568 | 0.0574 | 0.0672 | 0.0451 | 0.0853 | 0.1219 | **0.1542** |
| | BLEU | 0.1233 | 0.0786 | 0.0678 | 0.2888 | 0.0460 | 0.0247 | 0.2723 | 0.1981 | **0.5799** |
| | BERTScore | 0.7318 | 0.7464 | 0.7440 | 0.7402 | 0.7516 | 0.7424 | **0.7990** | 0.7827 | 0.7957 |

# 4 EXPERIMENTS

## 4.1 EXPERIMENTAL SETTINGS

**Datasets.** We conduct experiments on three benchmark datasets from Amazon (Hou et al., 2024)[1], including: *Book Review*, *Movie Review*, and *CD Review*, which have been widely used in recent studies on personalized text generation (Qiu et al., 2025b). The task involves generating personalized product reviews that reflect the user's preferences. Each user–item interaction contains the item title, item description, user rating, and review title, providing rich signals for personalization. In addition, we include the *Topic Writing* dataset (Kumar et al., 2024), where the task is to generate a personalized long-form Reddit post on a given topic based on a user-written summary. For each user, we construct a temporally ordered sequence of past interactions as user history. To balance sufficient historical context with the computational cost of long-sequence processing, we follow prior work (Qiu et al., 2025b) and cap the input length at 4096 tokens, matching the maximum context window of our backbone LLM.

**Baselines.** We compare our method against a diverse set of strong baselines spanning three categories. *(1) Base Model:* **Qwen** (Yang et al., 2024) is a strong LLM backbone without any personalization. *(2) Retrieval-based Methods:* **Contriever** (Lei et al., 2023) is a widely used dense retriever model, retrieve the most relevant past user interactions for instruction. **LatestK** (Liu et al., 2024a) selects the latest $K$ user interactions based on timestamp, assuming recency reflects more preference. **CoS** (He et al., 2024) amplifies the influence of context through controlling the activation during decoding. **LLM-TRSR** (Zheng et al., 2024) uses recurrent summarization to compress user history into a structured representation. *(3) PEFT-based Methods:* **SFT** (Hu et al., 2022) is the Standard supervised fine-tuning with the task-specific dataset. **ContextSFT** (Salemi et al., 2024b) is

---

[1]https://amazon-reviews-2023.github.io/

Table 2: Ablation results on personalized text generation across three benchmark datasets. "RI" (%) refers to the relative improvement of each ablation variant over the Base Model.

| Datasets (→) | Book Review | | | | Movie Review | | | | CD Review | | | |
|---|---|---|---|---|---|---|---|---|---|---|---|---|
| Methods (↓) | ROUGE-1 | RI | ROUGE-L | RI | ROUGE-1 | RI | ROUGE-L | RI | ROUGE-1 | RI | ROUGE-L | RI |
| **Base Model (SFT)** | 0.0752 | - | 0.0351 | - | 0.0620 | - | 0.0305 | - | 0.0668 | - | 0.0314 | - |
| **+ MCEOnly** | 0.1827 | 142.9 ↑ | 0.0907 | 158.4 ↑ | 0.1629 | 162.7 ↑ | 0.0796 | 161.0 ↑ | 0.1552 | 132.3 ↑ | 0.0744 | 136.9 ↑ |
| **+ MCE-DCE Alignment** | 0.1876 | 149.5 ↑ | 0.0961 | 173.6 ↑ | 0.1671 | 169.5 ↑ | 0.0817 | 168.0 ↑ | 0.1672 | 150.3 ↑ | 0.0793 | 152.5 ↑ |
| **+ DCE_Only** | 0.1958 | 160.4 ↑ | 0.1122 | 219.7 ↑ | 0.1865 | 200.8 ↑ | 0.0953 | 212.4 ↑ | 0.1805 | 170.2 ↑ | 0.0922 | 193.6 ↑ |
| **+ Full (NextQuill)** | **0.2318** | 208.2 ↑ | **0.1270** | 261.8 ↑ | **0.2015** | 225.0 ↑ | **0.1041** | 241.3 ↑ | **0.1976** | 195.8 ↑ | **0.0998** | 217.8 ↑ |

a supervised fine-tuning method that directly trains the LLM using history-augmented information. **OPPU** (Tan et al., 2025) is a PEFT method that learns lightweight user-specific adapters to encode preference signals. Notably, OPPU requires training a separate model for each individual user, unlike all other baselines. This design provides an upper bound for personalization performance but incurs prohibitive computational costs.

**Evaluation Metrics.** To evaluate the quality of generated reviews, we adopt a comprehensive set of metrics widely used in text generation. We report *ROUGE* (Lin, 2004), *METEOR* (Banerjee & Lavie, 2005), and *BLEU* (Post, 2018) to measure lexical and semantic overlap with ground-truth reviews. In addition, we employ *BERTScore* (Zhang et al., 2020) to assess semantic similarity in the embedding space. Finally, following recent advances in automatic evaluation, we employ an *LLM-as-a-judge* approach (Yang et al., 2024), where `Qwen2.5-72B-Instruct-AWQ` is prompted to provide holistic judgments (LLMScore) on both content relevance and stylistic alignment. The detailed results of this evaluation are presented in Appendix C.7, offering a more comprehensive perspective on personalized text generation quality.

**Implementation Details.** We use Qwen2.5-3B (Yang et al., 2024) as the backbone LLM for all methods. For PEFT Methods, we adopt low-rank adaptation (LoRA) (Hu et al., 2022) to train our models. We use the AdamW (Loshchilov & Hutter, 2017) optimizer with a learning rate of $5 \times 10^{-6}$, a weight decay of 0.025, and a dropout rate of 0.05. Besides, we employ Deepspeed (Rasley et al., 2020) for acceleration and ZeRO (Rajbhandari et al., 2020) for optimization. The train epochs is set to 5. For our framework, the threshold $\delta$ is set to 0.05. The parameter $\alpha$, which controls the weight of the personalized loss, is tuned in the range {0.01, 0.05, 0.1}. The high weight $\lambda$ is searched in {0.9, 0.8, 0.7}, while the low weight $\epsilon$ is set to 0.1.

## 4.2 OVERALL PERFORMANCE

We compare NextQuill with a wide range of baselines and report the results in Table 1. Across all datasets and evaluation metrics, NextQuill consistently achieves the best performance, demonstrating the effectiveness of our causal preference modeling framework in capturing and leveraging user-specific information. Notably, methods that incorporate user information (e.g., CoS, LLM-TRSR) significantly outperform those that do not (e.g., Qwen), confirming the critical role of user history in improving generation quality. Moreover, PEFT-based methods generally surpass retrieval-only approaches, suggesting that directly optimizing user-conditioned representations is more effective than simply injecting retrieved content. These findings validate our core design: modeling both the *data-side causal path* for ground-truth generation and the *model-side causal path* for prediction leads to stronger performance and more effective personalization.

## 4.3 IN-DEPTH ANALYSIS

**Ablation Studies.** To better understand the contribution of each component in our framework, we conduct an ablation study focusing on two key design choices: *causal preference modeling* and *token-level preference weighting*. We construct several ablation variants by selectively disabling these components (see Appendix B.2 for implementation details).

As shown in Table 2, incorporating DCE (DCEONLY), MCE (MCEONLY), or MCE-DCE alignment (MCE-DCE ALIGNMENT) results in significant improvements over SFT baseline, demonstrating that

Figure 2: Effect of hyper-parameter $\alpha$ on the performance.

Table 3: Comparison of ContextSFT and NextQuill across model sizes (1.5B, 3B, 7B), highlighting the robustness of NextQuill to model scale.

| Datasets | Metrics ($\downarrow$) | 1.5B | | 3B | | 7B | |
|---|---|---|---|---|---|---|---|
| | | ContextSFT | NextQuill | ContextSFT | NextQuill | ContextSFT | NextQuill |
| **Book Review** | ROUGE-1 | 0.1434 | 0.1631 | 0.1661 | 0.2318 | 0.3879 | **0.4116** |
| | ROUGE-L | 0.0728 | 0.0938 | 0.0836 | 0.1270 | 0.2183 | **0.2458** |
| | METEOR | 0.1734 | 0.2282 | 0.2158 | 0.2580 | 0.3317 | **0.3737** |
| | BLEU | 2.4144 | 2.7469 | 2.1214 | 3.5718 | 14.883 | **18.158** |
| **Movie Review** | ROUGE-1 | 0.1151 | 0.1251 | 0.1573 | 0.2015 | 0.3143 | **0.3246** |
| | ROUGE-L | 0.0547 | 0.0637 | 0.0753 | 0.1041 | 0.1526 | **0.1642** |
| | METEOR | 0.1286 | 0.1512 | 0.1718 | 0.1918 | 0.2477 | **0.2602** |
| | BLEU | 1.5693 | 1.6808 | 1.7151 | 2.3845 | 7.8608 | **9.1527** |
| **CD Review** | ROUGE-1 | 0.1171 | 0.1326 | 0.1505 | 0.1976 | 0.3189 | **0.3238** |
| | ROUGE-L | 0.0561 | 0.0653 | 0.0714 | 0.0998 | 0.1510 | **0.1576** |
| | METEOR | 0.1173 | 0.1547 | 0.1583 | 0.1805 | 0.2400 | **0.2542** |
| | BLEU | 1.1502 | 1.4181 | 1.3487 | 1.9676 | 7.2476 | **9.1314** |

each causal component independently enhances personalization. The full model (NextQuill), which integrates both components, achieves the best overall performance. These findings validate the effectiveness of our design: modeling both *what* the model learns from user history (via causal preference) and *where* to focus supervision (via token weighting) is crucial for enhancing personalization.

**Scaling Analysis.** Table 3 compares NextQuill with the strongest baseline ContextSFT across different backbone sizes (Qwen2.5-1.5B, 3B, 7B). We observe that NextQuill consistently outperforms ContextSFT on all datasets and metrics, with the performance gap widening as the model size increases. These results demonstrate that our causal preference alignment mechanism scales favorably with larger models, effectively leveraging additional capacity for improved personalization.

**Hyper-parameter Analysis.** We conduct experiments on the *CD Review* dataset to analyze the impact of two key hyper-parameters in our framework: (1) $\alpha$ in Equation 8, which controls the weight of the causal preference loss, and (2) $\lambda$ in Equation 4, which modulates the strength of the token-level preference weighting. To isolate their individual effects, we disable the token weighting strategy when tuning $\alpha$, and remove the weighted causal preference loss when tuning $\lambda$. The results for $\alpha$, evaluated across multiple metrics, are shown in Figure 2. We find that $\alpha = 0.05$ consistently yields strong performance and serves as a robust setting across datasets. The detailed analysis of $\lambda$ is reported in Appendix C.5.

### 4.4 WORD-LEVEL PREFERENCE ANALYSIS.

To further validate the ability of NextQuill to capture user preferences, we conduct an quantitative analyse at the token level. Specifically, we randomly sample 50 training instances from each dataset and compare the difference in token-level logits with and without user history for both NextQuill and ContextSFT. As shown in Figure 3, NextQuill produces significantly larger logit differences, indicating that it learns stronger preference representations. This confirms that our model captures more user-specific signals, which in turn helps better align the generated output with personalized aspects of the target text.

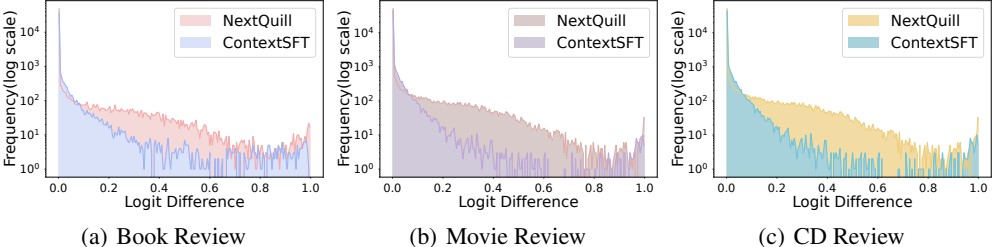

Figure 3: Logit difference distributions with and without user history across three benchmark datasets.

## 5 RELATED WORK

**LLM Personalization.** LLM personalization has gained increasing attention across text (Tseng et al., 2024; Zhao et al., 2025a; Zhang et al., 2025), conversation (Li et al., 2025; Zhao et al., 2025c) and multimodal generation (Shen et al., 2024). Recent Benchmarks such as LaMP (Salemi et al., 2024c) and LongLaMP (Kumar et al., 2024) provide standardized settings for training and evaluating personalized text generation. Existing methods mainly follow two paradigms: retrieval-based prompting and fine-tuning with user history. Retrieval-based approaches (Salemi et al., 2024a; Zhuang et al., 2024) augment model inputs by retrieving user-specific context from external memory. DPL (Qiu et al., 2025b) identifies inter-user differences to improve personalization. SteerX (Zhao et al., 2025b) leverages causal inference to disentangle preference-driven from preference-agnostic tokens and uses these signals to steer generation. Fine-tuning-based approaches adapt model parameters using user histories. Parameter-efficient methods (Tan et al., 2024a;b) achieve scalable personalization across users. ONCE (Liu et al., 2024c) constructs profile summaries from behavioral logs, and PPlug (Liu et al., 2024b) encodes user histories into personalized embeddings. P-RLHF (Li et al., 2024) jointly learns a user model and a personalized LLM from human feedback. DEP (Qiu et al., 2025a) further models inter-user differences in latent space rather than via prompt signals. However, most existing methods treat all tokens uniformly, lacking mechanisms to identify which parts of the input or output truly reflect user intent. In contrast, we introduce a causal preference modeling framework that attributes token-level effects through counterfactual reasoning, enabling principled supervision over both model-internal representations and output behavior to achieve genuine personalization.

**Causal Inference for LLMs.** Causal inference provides a principled framework for modeling cause-effect relationships in both observational and interventional settings (Peters et al., 2017). With the rise of LLMs, causal methods have been increasingly adopted to better understand, analyze, and improve model behavior across a range of NLP tasks (Stolfo et al., 2023; Zhu et al., 2024; Wang et al., 2023). A growing body of research investigates whether LLMs possess causal reasoning abilities (Jin et al., 2024), and explores how causal inference can be applied when treating text as either the treatment or the outcome variable (Zhou et al., 2024; Veitch et al., 2020). Other work leverages causal reasoning to improve LLM capabilities in domains such as recommendation systems (Zhang et al., 2021; 2023b) and arithmetic reasoning (Chi et al., 2024; Berg-Kirkpatrick & Spokoyny, 2020; Tseng et al., 2024). Despite these advances, causal techniques have not yet been systematically applied to the challenge of LLM personalization. To our knowledge, this work is the first to propose a unified causal framework that models user preference effects from both the model side and the data side, enabling fine-grained attribution and alignment for personalized text generation.

## 6 CONCLUSION

We introduce NextQuill, a novel causal preference modeling framework for LLM personalization that attributes and aligns user-specific signals through token-level causal effect estimation. By explicitly modeling both model-side and data-side preference pathways, NextQuill enables targeted, interpretable supervision aligned with true user preferences. Unlike prior methods that uniformly fit all predictions and target tokens, our approach aligns model-internal causal effects with preference-bearing supervision signals, allowing the model to learn *what* to adapt and *where* to focus. Experiments across diverse benchmarks confirm the effectiveness of NextQuill in improving personalized generation performance. We provide a detailed discussion of the limitations in Appendix D.

ETHICS STATEMENT

This research adheres to the ICLR Code of Ethics. It does not involve human subjects or sensitive personal information. We use only user data that individuals are willing to publicly disclose, *e.g.*, reviews or posts. Moreover, all user information is anonymized, containing no personally identifiable data, and our use of the dataset strictly adheres to the guidelines established by the data providers. Although our experiments and method design do not pose direct risks of harm, we acknowledge that applying our method in private domains should involve stronger privacy safeguards, such as providing users with control over their data and supporting local learning scenarios. Details regarding the use of large language models (LLMs) are provided in Appendix F.

REPRODUCIBILITY STATEMENT

To ensure reproducibility of our work, we provide detailed descriptions of our method, model architectures, and training procedures in Sections 3 and 4 of the main text. All hyperparameters, data preprocessing steps, and evaluation protocols are documented in Section 4.1 and Appendix B. Additionally, we include pseudo-code for our algorithm in Appendix C.8 and provide links to our code and datasets (https://github.com/juntaoyou/NextQuill).

ACKNOWLEDGMENTS

This work is supported by grants from the Research Grants Council of the Hong Kong Special Administrative Region, China (No. CUHK 14206625).

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

# A    DETAILED CAUSAL GRAPHS AND CAUSAL PREFERENCE EFFECTS

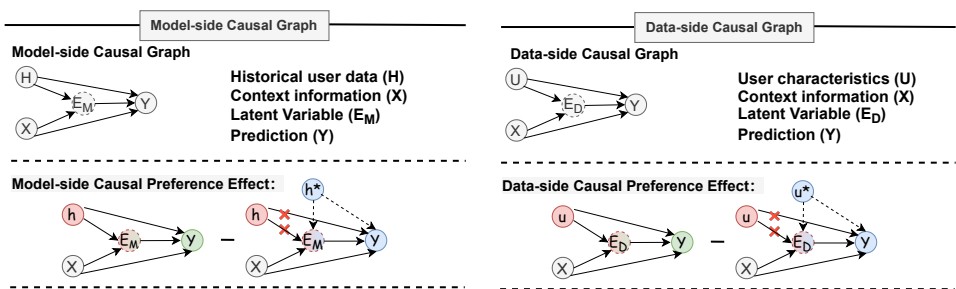

Figure 4: Causal graphs and corresponding preference effects. The left panel shows the model-side causal graph, and the right panel shows the data-side causal graph.

Figure 4 presents the causal graphs and corresponding preference effects considered in our framework. On the model side, we represent how historical user data ($H$) and context information ($X$) influence predictions ($Y$) through a latent variable $E_M$. The model-side causal effect isolates the contribution of user history by intervening on $H$ while holding other factors fixed. This quantifies the extent to which the model relies on historical data when generating personalized responses.

On the data side, user characteristics ($U$) interact with context information ($X$) through a latent variable $E_D$ to determine the ground-truth response $Y$. The data-side causal effect captures the impact of user-specific attributes by intervening on $U$, thereby reflecting how much personalization is encoded in the actual user-written reviews.

By jointly considering model-side and data-side causal preference effects, our framework establishes a principled way to better align model predictions with user preferences. This dual-view analysis underpins the design of our Causal Preference Alignment loss, which explicitly bridges the gap between how the model leverages history and how personalization manifests in user data.

# B    IMPLEMENTATION DETAILS

## B.1    COMPUTE RESOURCES

All experiments are conducted on NVIDIA A100 SXM4 GPUs with 84GB of GPU memory. We train our method on 4 A100 GPUs using mixed-precision training for approximately 26 hours per dataset. The training includes causal effect estimation via both factual and counterfactual forward passes, which introduces moderate additional overhead compared to standard fine-tuning. However, NextQuill requires no extra computation during inference: it performs a single forward pass without any retrieval, history reconstruction, or dynamic adaptation steps. This makes it both efficient and scalable at test time, offering lower inference overhead than current baselines that depend on retrieval or summarization mechanisms.

## B.2    ABLATION VARIANT IMPLEMENTATION DETAILS

To isolate the impact of each component in our framework, we evaluate the following ablation variants:

- **Base Model (SFT)**: The standard supervised fine-tuning on a LLM without any personalization mechanism. This is equivalent to disables both model-side causal preference effect (MCE) and data-side causal preference effect (DCE), and the alignment between MCE and DCE.

- **+ MCEOnly**: Add model-side causal preference effect (MCE), which use uniform weights by setting $\omega_t = 1$ in both the preference-weighted next token loss and causal preference alignment loss terms (Equations 6 and 7).

- **+ MCE-DCE Alignment**: Add the MCE and DCE alignment, which use uniform weights by setting $\omega_t = 1$ in preference-weighted next token loss (Equations 6) and use the causal preference alignment loss terms (Equations 7 ).
- **+ DCEOnly**: Add data-side causal preference effect (DCE), which only use the preference-weighted next token loss (Equations 6 ).
- **+ Full (NextQuill)**: Adds both preference-weighted next token loss and causal preference alignment loss, forming the complete NextQuill framework.

## C EXTENDED EXPERIMENTAL ANALYSIS

### C.1 NEGATIVE-CONTROL VALIDATION OF PREFERENCE-DRIVEN TOKEN IDENTIFICATION

| Dataset | Method | ROUGE-1 | ROUGE-L | METEOR | BLEU | BERTScore |
|---|---|---|---|---|---|---|
| Book Review | NextQuill-RW | 0.1811 | 0.0909 | 0.2170 | 2.2684 | 0.8024 |
| | NextQuill | **0.2308** | **0.1274** | **0.2573** | **3.3798** | **0.8182** |
| Movie Review | NextQuill-RW | 0.1569 | 0.0770 | 0.1628 | 1.8237 | 0.7947 |
| | NextQuill | **0.2009** | **0.1032** | **0.1909** | **2.2701** | **0.8064** |
| CD Review | NextQuill-RW | 0.1516 | 0.0730 | 0.1492 | 1.4167 | 0.7878 |
| | NextQuill | **0.1916** | **0.0973** | **0.1740** | **1.7313** | **0.8060** |

Table 4: Negative-control experiment where preference-driven tokens are replaced with randomly selected tokens (**NextQuill-RW**).

To further validate that the detected preference-driven tokens, we perform a negative-control test. Specifically, we construct a variant of our method, NextQuill-RandomWeighting, where the preference-driven token identification is replaced by randomly sampled tokens of the same quantity, while keeping the weighting mechanism unchanged. This variant intentionally destroys the causal preference signal, serving as a principled negative control.

As shown in Table 4, replacing the tokens identified by our causal preference estimator with random tokens results in consistent and substantial performance drops across all datasets and metrics. These results confirm that the causal preference effects estimated by our method are essential for personalization, and that the improvements of NextQuill are not due to trivial weighting heuristics.

### C.2 ROBUSTNESS TO POTENTIAL CONFOUNDERS: ITEM POPULARITY STRATIFICATION

| Datasets | Metrics | ContextSFT-LowPop | NextQuill-LowPop | ContextSFT-HighPop | NextQuill-HighPop |
|---|---|---|---|---|---|
| **Book Review** | ROUGE-1 | 0.1651 | **0.2079** | 0.1834 | **0.2336** |
| | ROUGE-L | 0.0826 | **0.1138** | 0.0919 | **0.1292** |
| | METEOR | 0.2149 | **0.2442** | 0.2406 | **0.2632** |
| | BLEU | 2.0779 | **3.0924** | 2.3964 | **3.5014** |
| | BERTScore | 0.8011 | **0.8114** | 0.8075 | **0.8187** |
| **Movie Review** | ROUGE-1 | 0.1588 | **0.1785** | 0.1748 | **0.1946** |
| | ROUGE-L | 0.0750 | **0.0939** | 0.0837 | **0.1009** |
| | METEOR | **0.1745** | 0.1724 | **0.1955** | 0.1845 |
| | BLEU | 1.6156 | **1.9791** | 1.9847 | **2.0724** |
| | BERTScore | 0.7956 | **0.8006** | 0.8036 | **0.8087** |
| **CD Review** | ROUGE-1 | 0.1531 | **0.1769** | 0.1682 | **0.1760** |
| | ROUGE-L | 0.0720 | **0.0905** | 0.0778 | **0.0905** |
| | METEOR | 0.1613 | **0.1650** | **0.1782** | 0.1639 |
| | BLEU | 1.3104 | **1.7201** | 1.5217 | **1.5433** |
| | BERTScore | 0.7909 | **0.7977** | 0.7982 | **0.8010** |

Table 5: Stratified analysis under potential confounder (item popularity). Items are divided into Low-Popularity and High-Popularity groups. NextQuill consistently outperforms ContextSFT across both groups, demonstrating robustness against popularity-related confounding effects.

To assess whether item popularity acts as a potential confounder in our personalization setting, we perform a stratified analysis based on item frequency. Following standard practices in recommen-

dation and personalization studies, items are divided into two groups—Low Popularity and High Popularity—and models are trained and evaluated separately on each subset.

We compare our method against the strongest baseline (ContextSFT) under both popularity regimes. As shown in Table 5, NextQuill consistently outperforms ContextSFT in both the Low and High popularity subsets, indicating that the gains of our causal preference modeling are not driven by popularity bias. These results demonstrate that NextQuill remains robust even when controlling for popularity-related confounding signals, strengthening the validity of our causal preference effects.

## C.3 RELIABILITY ANALYSIS OF THE DCE ESTIMATOR $f_{\theta_D}$

| Datasets | Metrics | ContextSFT-3B | NextQuill-1.5B (dynamic DCE) | NextQuill-3B (dynamic DCE) | NextQuill-3B (static DCE from Qwen2.5-3B) | NextQuill-3B (static DCE from ContextSFT-3B) | NextQuill-3B (static DCE from NextQuill-1.5B) |
|---|---|---|---|---|---|---|---|
| **Book Review** | ROUGE-1 | 0.1661 | 0.1631 | 0.2318 | 0.1907 | 0.2284 | 0.2549 |
| | ROUGE-L | 0.0836 | 0.0938 | 0.1270 | 0.1005 | 0.1237 | 0.1432 |
| | METEOR | 0.2158 | 0.2282 | 0.2580 | 0.2447 | 0.2917 | 0.2892 |
| | BLEU | 2.1214 | 2.7469 | 3.5718 | 2.8505 | 3.7309 | 4.3349 |
| **Movie Review** | ROUGE-1 | 0.1573 | 0.1251 | 0.2015 | 0.1635 | 0.1942 | 0.2107 |
| | ROUGE-L | 0.0753 | 0.0637 | 0.1041 | 0.0809 | 0.0938 | 0.1070 |
| | METEOR | 0.1718 | 0.1512 | 0.1918 | 0.1821 | 0.2094 | 0.2085 |
| | BLEU | 1.7151 | 1.6808 | 2.3845 | 1.9431 | 2.6177 | 2.6761 |
| **CD Review** | ROUGE-1 | 0.1505 | 0.1326 | 0.1976 | 0.1591 | 0.1931 | 0.2052 |
| | ROUGE-L | 0.0714 | 0.0653 | 0.0998 | 0.0762 | 0.0913 | 0.1019 |
| | METEOR | 0.1583 | 0.1547 | 0.1805 | 0.1722 | 0.1980 | 0.1954 |
| | BLEU | 1.3487 | 1.4181 | 1.9676 | 1.5286 | 2.1880 | 2.2284 |

Table 6: Comparison of different implementations of the DCE estimator $f_{\theta_D}$ across datasets.

To assess whether the DCE estimator $f_{\theta_D}$ serves as a reliable approximator for the causal preference effect in Equation 5, we compare several implementations of the estimator that differ in model size and training strategy. Specifically, we examine four variants: a 3B supervised baseline (ContextSFT-3B), a 1.5B NextQuill model trained with dynamic DCE estimation (NextQuill 1.5B, dynamic), a 3B NextQuill model using dynamic DCE estimation (NextQuill 3B, dynamic), and a 3B NextQuill model trained with static DCE weights computed by the 1.5B model (NextQuill 3B, static-from-1.5B). This analysis evaluates whether a weaker estimator compromises the fidelity of DCE computation.

As shown in Table 6, across the Book Review, Movie Review, and CD Review datasets, the 3B dynamic variant achieves better performance than the strongest baseline ContextSFT-3B, indicating that dynamically updating the estimator yields substantial gains. Additionally, the static-from-1.5B variant outperforms the 1.5B dynamic model, demonstrating that the 1.5B estimator is sufficiently strong to produce meaningful DCE weights. We consider both approaches reasonable; however, the static-from-1.5B method involves additional training cost due to precomputing weights, so we adopt the dynamic DCE strategy as the default in NextQuill.

## C.4 EXTENDED WORD-LEVEL PREFERENCE ANALYSIS.

*For the qualitative analysis*, we collect the tokens weighted by our method and perform K-Means clustering based on their final-layer hidden states. The resulting clusters, visualized in Figure 5, reveal clear separation among personalized tokens, common tokens, and auxiliary tokens found in user reviews, suggesting that our strategy effectively identifies and organizes tokens by their relevance to personalization.

## C.5 HYPER-PARAMETER ANALYSIS OF $\lambda$

We conduct a hyperparameter analysis by tuning $\lambda$ over the range $\{0.7, 0.8, 0.9\}$ to investigate its effect on model performance. As shown in Figure 6, we observe a consistent improvement across all evaluation metrics as $\lambda$ increases. This suggests that assigning higher weight to preference-bearing tokens helps the LLM better capture user-specific signals, thereby improving the quality of personalized text generation.

## C.6 ANALYSIS OF MODEL PERFORMANCE ACROSS DIFFERENT USER HISTORY LENGTHS

Table 7 presents the model's performance under two history lengths (1024 and 2048 tokens), as shown in Table XX. The results reveal two consistent trends. First, personalization quality improves

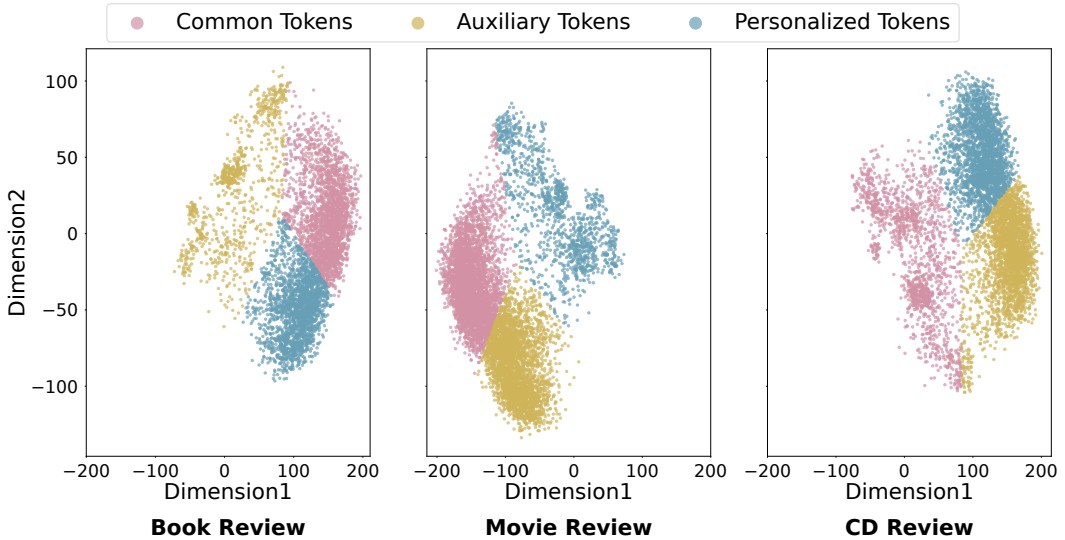

Figure 5: Visualization of token clustering based on hidden representations on three datasets.

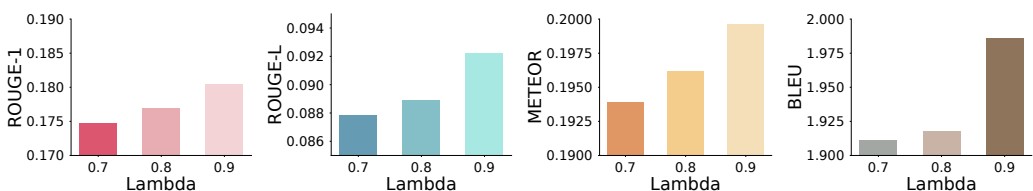

Figure 6: Effect of hyper-parameter $\lambda$ on the performance.

Table 7: Results under difference history lengths.

| Datasets (→) | Book Review | | | | Movie Review | | | | CD Review | | | |
|---|---|---|---|---|---|---|---|---|---|---|---|---|
| Methods (↓) | ROUGE-1 | ROUGE-L | METEOR | BLEU | ROUGE-1 | ROUGE-L | METEOR | BLEU | ROUGE-1 | ROUGE-L | METEOR | BLEU |
| ContextSFT | 0.1661 | 0.0836 | 0.2158 | 2.1214 | 0.1573 | 0.0753 | 0.1718 | 1.7151 | 0.1505 | 0.0714 | 0.1583 | 1.3487 |
| Length = 1024 | 0.1978 | 0.1071 | 0.1859 | 3.0821 | 0.1888 | 0.0977 | 0.1598 | 1.9795 | 0.1870 | 0.0938 | 0.1540 | 1.6705 |
| Length = 2048 | 0.2318 | 0.1270 | 0.2580 | 3.5718 | 0.2015 | 0.1041 | 0.1918 | 2.3845 | 0.1976 | 0.0998 | 0.1805 | 1.9676 |

as more user history becomes available: longer histories provide richer preference signals, leading to higher scores across ROUGE, METEOR, and BLEU. Second, the model remains robust in short-history scenarios, outperforming the strongest baseline (ContextSFT) even when the available history is limited. These findings confirm that our method effectively leverages user information while maintaining strong performance under history-limited settings.

## C.7 COMPREHENSIVE RESULTS VIA LLM-AS-A-JUDGE EVALUATION

Table 8: Main results of our NextQuill method across four benchmark datasets, evaluated using the LLM-as-a-judge paradigm. The reported metric is *LLMScore*, obtained by prompting `Qwen2.5-72B-Instruct-AWQ` to assess both content relevance and stylistic alignment. **Bold** numbers indicate the best performance, while underlined numbers indicate the second-best. We also report the relative improvement (%) over the base model Qwen.

| Category | Methods | Book Review | | Movie Review | | CD Review | | Topic Writing | |
|---|---|---|---|---|---|---|---|---|
| | | Score | Δ (%) | Score | Δ (%) | Score | Δ (%) | Score | Δ (%) |
| Base Model | Qwen | 1.0725 | – | 1.0260 | – | 1.0382 | – | 1.0012 | – |
| Retrieval-based | Contriever | 1.3513 | +26.0 | 1.2682 | +23.6 | 1.2972 | +24.9 | 1.4080 | +40.6 |
| | LatestK | 1.3544 | +26.3 | 1.2284 | +19.7 | 1.2834 | +23.6 | 1.3654 | +36.4 |
| | LLM-TRSR | 1.5236 | +42.1 | 1.4545 | +41.8 | 1.6977 | +63.5 | 1.4237 | +42.2 |
| | CoS | 1.1987 | +11.8 | 1.1032 | +7.5 | 1.1334 | +9.2 | 1.2263 | +22.5 |
| PEFT-based | SFT | 1.1041 | +2.9 | 1.0452 | +1.9 | 1.0895 | +4.9 | 0.9971 | -0.01 |
| | ContextSFT | 1.7911 | +67.0 | 1.4664 | +42.9 | 1.6157 | +55.8 | 1.6966 | +69.4 |
| | NextQuill | **2.1487** | +100.3 | **1.6984** | +65.5 | **1.9267** | +85.6 | **2.0506** | +104.8 |

---

**Prompt C.1: LLM-as-a-judge Evaluation Prompt (On Book Review, Movie Review, CD Review Datasets)**

You are a helpful assistant. Please act as an impartial judge and evaluate the quality of the response to instruction of the user displayed below. Based on the scoring criteria, please provide a score to the response compared to the reference and an explanation that why you assign the score to the response. Be as objective as possible. You should consider both content and writing style similarity to assign a score.

Your inputs:
instruction: the instruction provided to the AI assistant. reference: the correct answer to the instruction. response: the response generated by the AI assistant.

Scoring Criteria: You should assign a score to the response based on the following criteria:
Score 0: The answer is completely unrelated to the reference.
Score 1: The answer has minor relevance but does not align with the reference.
Score 2: The answer has moderate relevance but contains inaccuracies.
Score 3: The answer aligns with the reference but has minor omissions.
Score 4: The answer is completely accurate and aligns perfectly with the reference.

instruction: <instruction> The user is currently interacting with the item titled {item_title}, which is described as {item_desc}. The rating given is {rating}, and the review title written is {title}. Please generate a personalized review for the user based on this information. </instruction>

reference: <reference>output</reference>
response: <response>response</response>

Please provide a score to the response and an explanation that why you assign the score to the response. Format your response as:
"Score: [number] \n Explanation: [your detailed reasoning]"

---

**Prompt C.2: LLM-as-a-judge Evaluation Prompt (On Topic Writing Dataset)**

You are a helpful assistant. Please act as an impartial judge and evaluate the quality of the response to instruction of the user displayed below. Based on the scoring criteria, please provide a score to the response compared to the reference and an explanation that why you assign the score to the response. Be as objective as possible. You should consider both content and writing style similarity to assign a score.

Your inputs:
instruction: the instruction provided to the AI assistant. reference: the correct answer to the instruction. response: the response generated by the AI assistant.

Scoring Criteria: You should assign a score to the response based on the following criteria:
Score 0: The answer is completely unrelated to the reference.
Score 1: The answer has minor relevance but does not align with the reference.
Score 2: The answer has moderate relevance but contains inaccuracies.
Score 3: The answer aligns with the reference but has minor omissions.
Score 4: The answer is completely accurate and aligns perfectly with the reference.

instruction: <instruction> The user is currently writing a passage of a topic. The summary of the topic is {summary}. </instruction>

reference: <reference>output</reference>
response: <response>response</response>

Please provide a score to the response and an explanation that why you assign the score to the response. Format your response as:
"Score: [number] \n Explanation: [your detailed reasoning]"

---

For LLM-based evaluation, we adapt the ExPerT method (Salemi et al., 2025), an explainable evaluation framework designed for personalized text generation. ExPerT evaluates alignment along key personalization attributes such as content and writing style by extracting fine-grained aspects from both the generated and reference texts. Prompts C.1 and C.2 show the evaluation prompts used for the review-generation and topic-writing tasks, respectively.

Table 8 summarizes the results of our experiments on personalized text generation. The following key observations emerge. First, retrieval-based methods achieve substantial improvements over the base Qwen model, thereby confirming the utility of leveraging user history. Second, PEFT-based approaches deliver more stable and robust improvements. Specifically, ContextSFT produces significant gains over both Qwen and retrieval-based methods, demonstrating that direct fine-tuning on history-augmented inputs effectively encodes personalization signals. Most notably, NextQuill consistently attains the best performance across all datasets, which underscores the efficacy of our preference modeling strategy in jointly capturing content relevance and stylistic alignment.

## C.8 DETAILED ALGORITHM DESCRIPTION

For completeness, we provide the pseudocode of our training algorithm in Algorithm 1. The procedure illustrates how NextQuill performs causal preference alignment for LLM personalization. Specifically, the algorithm iterates over each training batch and sample, computes both model-side and data-side causal effects, and assigns preference weights to tokens accordingly. These weights are then used to optimize two complementary objectives: the preference-weighted next-token loss and the causal alignment loss. The final model parameters are obtained by jointly minimizing these losses. It is important to note that our method introduces additional computation only during training. At inference time, NextQuill does not incur any extra overhead compared to the base LLM, making it efficient and directly applicable in real-world personalized generation scenarios.

---

**Algorithm 1** NextQuill: Causal Preference Alignment for LLM Personalization

---

**Require:** Dataset $\mathbb{D} = \{(x, h, y)\}$, pre-trained LLM $f_\theta$, thresholds $\delta$, weights $\lambda, \epsilon$, alignment coefficient $\alpha$, learning rate $\eta$
**Ensure:** Fine-tuned model parameters $\theta^\star$

1: **for** each training batch from $\mathbb{D}$ **do**                   ▷ Iterate over training batches
2:     **for** each sample $(x, h, y)$ in batch **do**              ▷ Iterate over samples in the batch
3:         **for** $t = 1$ to $|y|$ **do**                              ▷ Iterate over each token
4:             $MCE_t \leftarrow f_\theta(x, h, y_{<t}) - f_\theta(x, \emptyset, y_{<t})$        ▷ Compute model-side causal effect
5:             $DCE_t \leftarrow f_{\theta_\mathbb{D}}(y_t | x, h, y_{<t}) - f_{\theta_\mathbb{D}}(y_t | x, \emptyset, y_{<t})$          ▷ Estimate data-side CE
6:             $\omega_t \leftarrow \begin{cases} \lambda, & DCE_t > \delta \\ \epsilon, & \text{otherwise} \end{cases}$            ▷ Assign preference weight to the token
7:         $L_n \leftarrow \frac{1}{|\text{batch}|} \sum_{(x,h,y)} \sum_{t=1}^{|y|} \omega_t \cdot \ell(f_\theta(x, h, y_{<t}), y_t)$       ▷ Compute preference-weighted next-token loss
8:         $L_p \leftarrow \frac{1}{|\text{batch}|} \sum_{(x,h,y)} \sum_{t=1}^{|y|} \omega_t \cdot \ell\big(f_\theta(x, h, y_{<t}) - f_\theta(x, \emptyset, y_{<t}); y_t\big)$       ▷ Compute causal alignment loss
9:         $\theta \leftarrow \theta - \eta \nabla_\theta(L_n + \alpha \cdot L_p)$                   ▷ Update model parameters
10: **return** $\theta^\star \leftarrow \theta$                              ▷ Return fine-tuned model parameters

---

## D    A DETAILED DISCUSSION ON THE LIMITATIONS

While NextQuill demonstrates strong improvements in personalization quality and interpretability, it also inherits several limitations that are common across existing personalization methods and highlight open challenges for future research.

- This work relies on user history to estimate meaningful causal effects. As with most behavior-driven personalization frameworks, the availability and quality of user interaction data directly impact the accuracy of preference attribution. In low-resource or cold-start scenarios, distinguishing true preferences from incidental patterns becomes more challenging. We view this as an opportunity to integrate auxiliary signals (e.g., demographic features, side information) to enhance robustness in sparse-data regimes.

- This method introduces additional training-time computation due to the need to evaluate both factual and counterfactual predictions per instance. Although this design enables more accurate supervision via causal attribution, it increases training overhead compared to standard fine-tuning. Importantly, this overhead is only related to training and does not affect inference speed, making the method practical at test time and more efficient than many baseline approaches that rely on repeated retrieval or dynamic adaptation during generation.

- Our work currently operates within a single-session context window and models personalization at the token level. While effective in capturing local preference signals, it does not explicitly model longer-term user dynamics or cross-session behaviors, which may be important in settings such as lifelong personalization, multi-turn dialogue, or recommendation-oriented generation.

- In this work, we construct the graph from domain-level knowledge: on the model side, it mirrors the prediction process, and on the data side, it captures how user responses are shaped by the user, the item, and their interaction. While alternative structures could also be considered, we provide further discussion in Appendix E.

## E    CAUSAL GRAPH SELECTION

For the structured causal model framework we used, the key is that the causal graph accurately reflects the mechanisms generating the relevant variables. In this work, we construct the causal graph for personalized generation based on domain-level common knowledge.

On the model side, the causal graph naturally represents the process by which the model generates predictions, providing an accurate characterization.

On the data side, we derive the causal graph in Figure 1 from domain knowledge. We find that user responses are primarily influenced by the user, the item, and their interaction. Based on this, we draw the causal graph and correspondingly design our method. The experimental results indirectly support the plausibility of this causal graph.

Some may argue that it is possible that other factors could affect user responses. The most notable is the "influence of other items," which in our scenario largely corresponds to items previously observed by the user. These are already accounted for as part of the user history. Some may think there are also "unobserved variables" that also have influences. This issue is beyond the scope of this work. However, if no unobserved confounders exist, our approach remains valid; if unobserved confounders are present, an extension may be necessary. In future work, we will extend our method to handle unknown confounders.

## F  LLM USAGE

We used large language models (LLMs) solely as a general-purpose writing aid to check grammar and improve the phrasing of certain sentences. LLMs were not involved in the research ideation, experimental design, data analysis, or any technical content creation. All scientific contributions, results, and interpretations in this work were generated solely by the authors.

## G  CASE STUDY

**Item Metadata:**
Item Title: The Sound of Light: (A World War II Resistance Book and Inspirational Christian Romance)
Item Description: From the Back Cover Copenhagen, 1940 When the Germans march into Denmark, Baron Henrik Ahlefeldt exchanges his nobility for anonymity, assuming a new identity so he can secretly row messages for the Danish Resistance across the waters to Sweden...
Output Review Rating: 4.0
Output Review Title: WWII historical romance

**User's Real Review:**
*The Sound of Light is a very well written WWII historical romance* I highly recommend this book to those who enjoy historical fiction. *My thanks to the publisher and Netgalley for my advance ebook*. *This is my unbiased review.*

**NextQuill Generated Review:**
*The Sound of Light is a well written WWII historical romance* I thoroughly enjoyed the plot and the character dynamics. Fans of WWII fiction will enjoy this book. *My thanks to the publisher and Netgalley for my advanced copy*. *This is my unbiased review.*

**ContextSFT Generated Review:**
I really loved this story! It started slow, which isn't always good thing, especially since we're talking about war here. However, after getting used to the pace, I found myself engrossed in the world created by Ms. Sundin. There were some parts where I felt like I needed more background info, however those moments didn't last long enough to make them annoying. Overall, I would definitely read again! *The main focus of the story is how else deals with being Jewish during WW2 while still maintaining her relationship with Henrik.* While dealing with all sorts of problems herself, she finds ways to help others too. Her relationships with various members of the community add depth to the overall picture presented by Ms. Sundin. As far as the romance goes, it's pretty tame compared to most books I've read lately. Still, it works perfectly fine within its context. All-in-all, I'd say this is a solid pick-up-and-read kind of book. Highly recommended !

Figure 7: Case study comparing personalized review generations from NextQuill and ContextSFT on the *Book Review* dataset. Sentences highlighted in color indicate alignment with the user's actual review, revealing the degree of preference capture in each method.

To complement the quantitative results, we further provide case studies to qualitatively illustrate how NextQuill captures user-specific preferences in generated text. We select representative samples from the Book Review, Movie Review, and CD Review datasets, and compare the reviews generated by NextQuill with those produced by our strong baseline ContextSFT. As shown in Figures 7, 8 and 9, we highlight sentences that align with the user's actual review using the same color.

Across all domains, the outputs from NextQuill exhibit stronger alignment with the user's real preferences, often reflecting personalized descriptions, opinions, or individual reading and viewing experiences. In contrast, reviews generated by ContextSFT tend to be generic, containing surface-level

**Item Metadata:**
Item Title: Stunt
Item Description: Barenaked Ladies follow the certified-Gold success of Rock Spectacle with their hotly anticipated new studio album – Stunt...
Output Review Rating: 5.0
Output Review Title: Easily the best Barenaked Ladies album I own!!

**User's Real Review:**
*This is the album that did it for me. "Stunt" is the album by now one of my favorite bands that really got me into Barenaked Ladies.* I have owned it for about 8 or so years now and these guys have been one of my favorite bands since. I've waited too long to write this review about my limitless praise. *I have NEVER EVER gotten tired of ANY of the excellent songs on this god-like album and I never will be.* I now own several of Barenaked Ladies' albums {I got Maroon not soon after this one} and even a DVD chock-full of wonderful music videos they've made over the years called "Barelaked Nadies."For me, this album is to soft rock music as Seinfeld is to sitcoms {except this album isn't about nothing, he he}. I've taken it along with me on a musical journey to quite a few state trips I've gone on {including NY and Washington DC. This cd is so dear to me I HAVE to take it everywhere}. If I had to recommend any cd that I never do get tired of and can listen to all day long all the way through if I chose, Stunt would be it. *Every song is totally great.* This is one of the most impressive alternative rock albums I've ever heard. Highly recommended if you like this kind of gem genre. No doubt you've heard at least a few of these songs on the radio. Hope you like it almost as much as I do.

**NextQuill Generated Review:**
This album has been released recently and is definitely worth buying. There are lots of great songs on this album, including " The Boys Are Back In Town", "Rock Spectacle", "Stunt", "One Week", "It's All Been Done", and many more. *These songs are all good and each one stands alone nicely.* However, the thing that makes this album stand out above the rest is the fact that every single song on this album is amazing!! Each song is unique and different from the next, but they all work perfectly together. *They don't seem forced or awkward, and everything fits together perfectly.* I am amazed at how talented these people are and how they manage to create such beautiful music. I cannot wait to see what else they come up with in the future. *This album is definitely a must-have for any Barenaked Ladies fan!!!*

**ContextSFT Generated Review:**
Bare Naked Ladies has been doing great things lately. They released their second album called "Rock Spectacle", which had lots of good songs including "The World Is Not Enough". *Their third album, "Stunt", is easily the best album I own!!* There are tons of fun songs on this album, ranging from the upbeat "Weirdo" to the sad "Tears In Heaven". *But overall, this is probably the best album I own!!!* Buy it today!!!

Figure 8: Case study comparing personalized review generations from NextQuill and ContextSFT on the *CD Review* dataset. Sentences highlighted in color indicate alignment with the user's actual review, revealing the degree of preference capture in each method.

**Item Metadata:**
Item Title: Hidden Figures
Item Description: HIDDEN FIGURES is the incredible untold story of Katherine G. Johnson (Taraji P. Henson), Dorothy Vaughan (Octavia Spencer) and Mary Jackson (Janelle Monáe)-brilliant African-American women working at NASA, who served as the brains behind one of the greatest operations in history...
Output Review Rating: 4.0
Output Review Title: Important Movie with Historical Significance; Highly Recommended

**User's Real Review:**
Not many movies come along these days that I can recommend without any hesitation. This is one. Based on a true story, which I always find interesting, it tells of the women; particularly the black women who performed much of the complicated math involved in getting our astronauts out into space and back again safely. *It shows the struggles they went through for acceptance. You NEED this movie in your collection, if for no other reason than to help complete your knowledge of unknown or little known history. As usual, with true stories, there were some liberties taken with the facts, but for the most part the story is true.* I only knocked off one star from a 5-star rating due to the fact that even though it was very entertaining, it wasn't one of those movies that if you didn't see it you would be missing out on something incredible. A wonderful movie that I highly recommend, but not an incredible one; very close to it, though. A 5+ rating for the historical significance."

**NextQuill Generated Review:**
*This is an important historical film that tells the true story of three brilliant African American mathematicians who played a key role in the space program during the Cold War era.* Their work helped pave the way for future astronauts and scientists. *The movie is based on the real-life stories of Katherine G. Johnson, Dorothy Vaughan, and Mary Jackson, who were instrumental in launching America's first satellite and later sending humans into space. The movie is highly recommended for those interested in history, science, and technology. It also provides insight into the struggles of black women in the workplace during the Civil Rights Movement.* The actors did a fantastic job portraying the characters and bringing their experiences to life. The script is excellent and keeps the audience engaged throughout the entire movie. Although there are some technical details that might confuse viewers, overall, this is a fascinating and informative movie that deserves high praise.

**ContextSFT Generated Review:**
*This is definitely a historical film based on true events.* It tells the amazing tale of three brilliant black mathematicians during World War II. These women did work essential to our country winning WWII. Their contributions went unnoticed until recently due to racism and sexism. We owe them a debt of gratitude! Taraji P. Henson gives us her best performance yet. Octavia Spencer also delivers a strong performance. Janelle Monae doesn't stand out enough here compared to the other actresses, although she still manages to deliver a fine performance. Overall, this is a highly enjoyable movie. A little slow paced towards the beginning, but once the action gets going, there aren't many dull moments left. You'll enjoy watching this movie. Definitely worth seeing!

Figure 9: Case study comparing personalized review generations from NextQuill and ContextSFT on the *Movie Review* dataset. Sentences highlighted in color indicate alignment with the user's actual review, revealing the degree of preference capture in each method.

compliments or templated statements with little user-specific relevance. These qualitative examples demonstrate that our framework can generalize across domains and user profiles, maintaining both coherence and personalization in text generation.

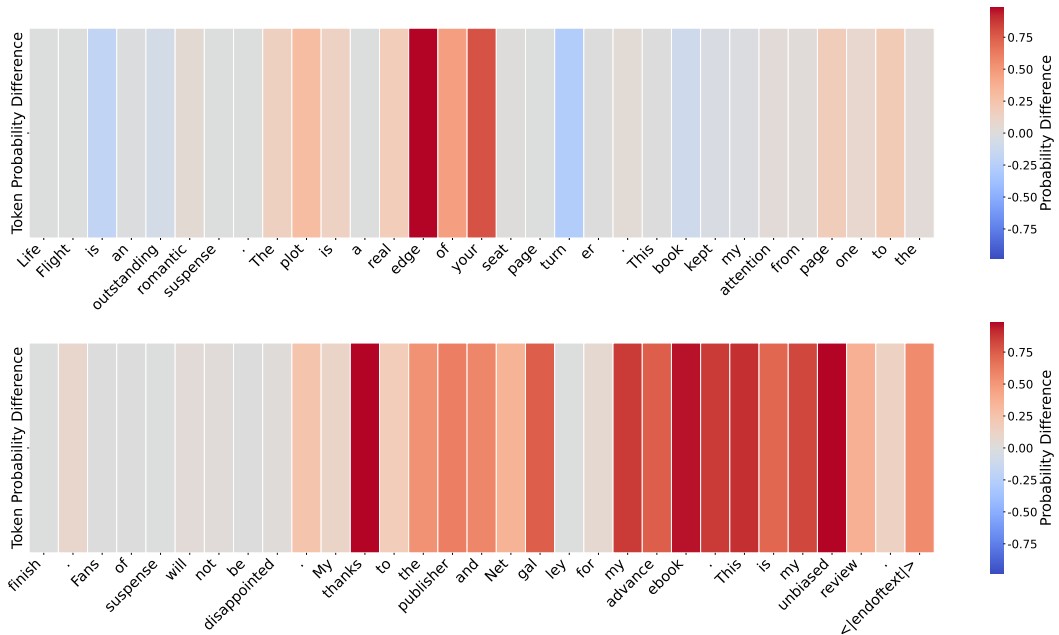

Figure 10: Heatmaps of token-level probability differences of NextQuill on *Book Review* dataset.

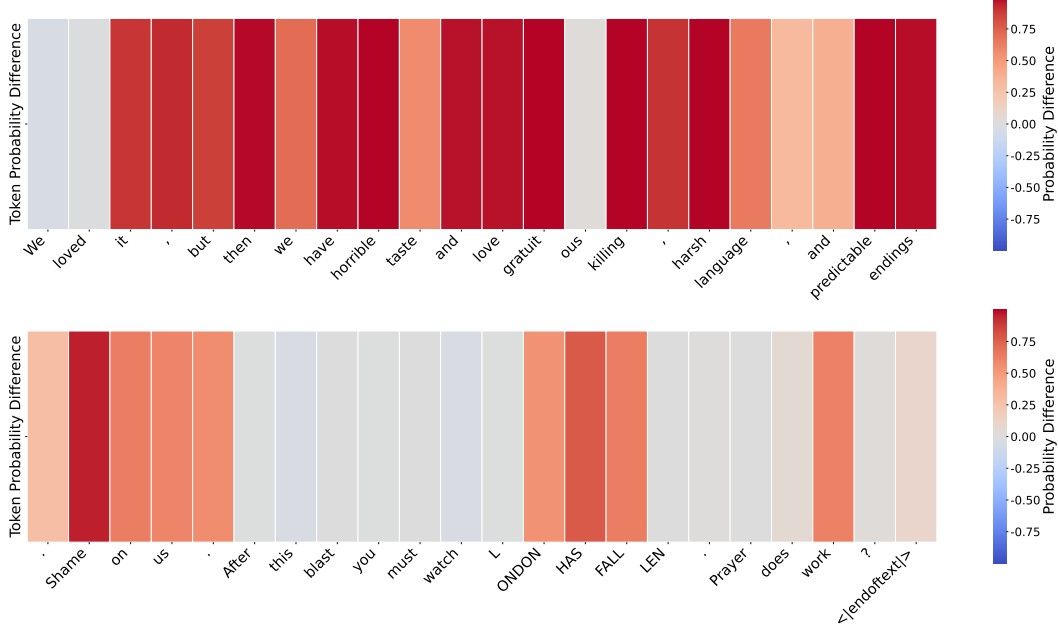

Figure 11: Heatmaps of token-level probability differences of NextQuill on *Movie Review* dataset.

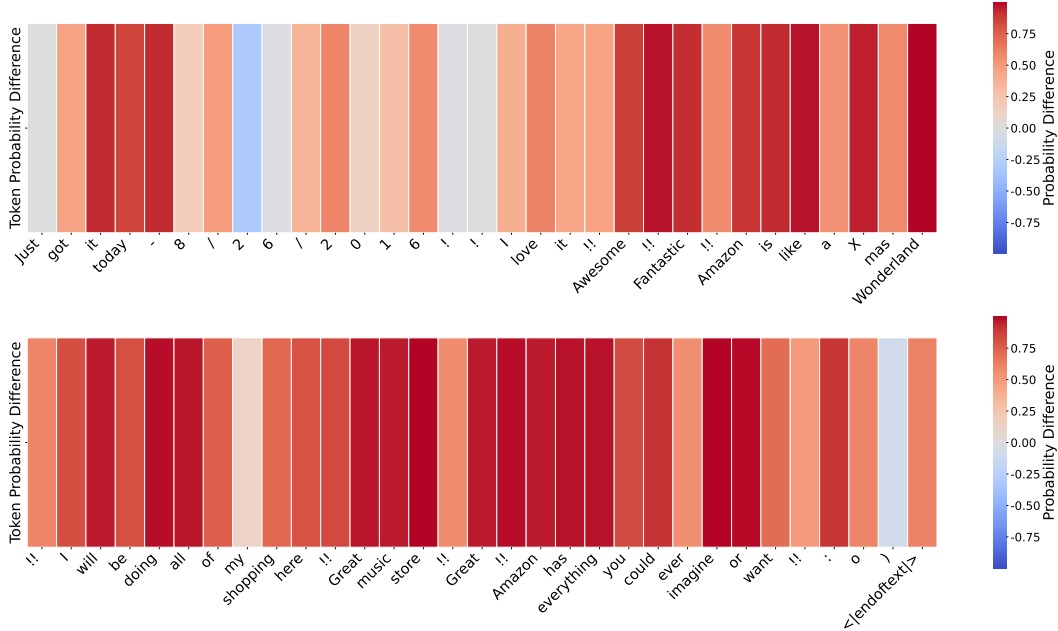

Figure 12: Heatmaps of token-level probability differences of NextQuill on *CD Review* dataset.

Furthermore, we include heatmap visualizations of token-level probability differences produced by NextQuill across all three datasets. As shown in Figures 10, 11, and 12, the DCE-based heatmaps separate preference-driven tokens from contextually relevant but preference-agnostic content. These consistent patterns demonstrate that our DCE mechanism accurately identifies user-preference signals, validating the effectiveness of our preference-driven token weighting strategy.

