# OpenReview forum: "NextQuill: Causal Preference Modeling for Enhancing LLM Personalization"
_ICLR.cc/2026/Conference — ICLR 2026 Poster_

### Official Review · Reviewer_y5Zw · 2025-10-29

**Soundness:** 3
**Presentation:** 3
**Contribution:** 3
**Rating:** 4
**Confidence:** 3

**Summary:**

The paper presents NextQuill, a new fine-tuning framework for LLM personalization. It argues that current methods are too “shallow” because they don’t separate true preference signals from generic contextual noise in training data. It takes a causal approach, distinguishing preference-driven from context-driven components in both model outputs and ground truth. It introduces two alignment strategies: (1) matching the model’s causal preference effects to real data, and (2) applying preference-weighted supervision that focuses on preference-driven tokens. The authors report improvements across several personalization benchmarks.

**Strengths:**

(1) **Clear problem design**: The paper clearly outlines the limitations of existing methods that treat all tokens equally and makes a compelling case for adopting a preference-aware learning approach.

(2) **Novelty**: The work introduces a novel approach to disentangle user preferences from contextual factors. Its dual-sided causal graphs, covering both the model and data sides, provide a clear and elegant theoretical foundation.

(3) **Strong empirical results**: Improvements are consistent across metrics (ROUGE, METEOR, BLEU, BERTScore) and scale with model size. It includes diverse baselines spanning retrieval-based and fine-tuning methods, and ablation studies highlight the importance of each framework component.

(4) **Interpretability**: The word-level preference analysis and case study offer qualitative insights that effectively support the paper’s contributions.

**Weaknesses:**

(1) **Loss inconsistency**: In the main text, $L_p$ in Eq. 7 aligns MCE to the ground truth label token $y_t$ via cross-entropy. However, in Algorithm 1 in the Appendix, $L_p$ aligns MCE directly to DCE, that is, effect-to-effect rather than effect-to-label. These are not equivalent objectives. If the intended goal is “align model-side and data-side effects,” the algorithmic version matches the claim most literally, but the equation in text uses the ground label token as the target. This critical inconsistency makes the paper's core method ambiguous and, as written, unreproducible for now.

(2) **Possible information leakage in DCE estimation**: DCE is estimated using the model  $f_{\theta_D}$,  which has been trained on the dataset D. This setup raises the possibility of information leakage, as $f_{\theta_D}$ may inadvertently encode knowledge of the data distribution. Consequently, the resulting DCE scores might not represent purely causal estimates but could instead be partially influenced by model memorization.

(3) **Limited Scope of Personalization**: As noted in the limitations, the approach focuses on single-session, token-level personalization without considering long-term user dynamics or cross-session behaviors, which limits applicability to real-world scenarios.

**Questions:**

(1) **$L_p$ clarification**: A precise and consistent description of the implemented $L_p$ formulation would strengthen the paper. Clearly delineating which version, effect-to-label or effect-to-effect, is used in experiments, and providing a detailed breakdown of the loss components, would help in resolving the ambiguity surrounding this objective.

(2) **DCE estimation methodology**: The current setup of DCE estimation could potentially lead to information leakage. An analysis or justification of why this doesn't introduce bias would be helpful. In case of potential leakage, considering or comparing alternative estimation strategies might be helpful.

(3) **Low-user interaction performance**: While the paper acknowledges that limited user history constrains personalization performance, it might be valuable to quantify this effect empirically. An analysis showing how model performance varies with different levels of user interaction (e.g., short vs. long histories) would clarify the method’s robustness in low-interaction or cold-start settings.

(4) Appendix G states that the method’s validity relies on the assumption of no unobserved confounders. A discussion of potential confounders in this context and how their presence might affect the proposed method would add important context to the paper’s causal claims.

---

> ### Author Response · Authors · 2025-11-22
> **Response to Reviewer y5Zw**
>
> Dear Reviewer y5Zw,
>
> Thank you for the helpful suggestions. Below we provide our detailed responses, and we hope they address your concerns.
>
> **W1 & Q1: Loss clarification: effect-to-label (Eq. 7) or effect-to-effect (Alg. 1)?**
>
> **A1:** Sorry for the inconsistency. The correct version is the effect-to-label formulation, which aligns with the training paradigm of LLMs. In practice, effect-to-label can be viewed as an approximation of effect-to-effect alignment: The effect-to-label objective is written as $\sum -w_t\log(\text{MCE}[y_t])$,
> where $(w_t = \lambda )$ if $(\text{DCE}[y_t] > \text{threshold})$ and $(w_t = \epsilon)$ otherwise, with $\lambda > \epsilon$. This design encourages tokens with higher DCE values to receive higher $\text{MCE}[y_t]$ during training.
>
> **W2 & Q2: The DCE Estimation Faces Potential Leakage**
>
> **A2:** Thanks for the question. There is no information leakage in our DCE estimation. First, the estimation is conducted solely on the training data and never accesses the validation or test sets. Second, the standard causal-effect estimation inherently relies on statistical quantities such as conditional probabilities of the observed data. This does not constitute information leakage; it is analogous to using training data to learn model parameters. Our estimation follows the same principle.
>
> **W3. Single-session, token-level personalization without considering long-term user dynamics or cross-session behaviors, limiting applicability to real-world scenarios.**
>
> **A3:** We thank the reviewer for this valuable comment. We clarify that while we acknowledge this limitation in the paper, it can be naturally addressed by combining our method with the standard "memory + retrieval" paradigm, which supports long-term and cross-session personalization by retrieving relevant parts from the full user history. In our implementation, most of methods (including ours) already include a retrieval process to identify the most relevant items from the full user history. However, in our datasets, we empirically find that using only the most recent history yields better performance.
>
> **Q3: How does model performance vary with different levels of user interaction (e.g., short vs. long history)?**
>
> **A3:** We have added results across different history lengths in the table below. The results show that (1) performance improves as user history becomes longer, and (2) the model consistently outperforms the best-performing baseline (ContextSFT) even in short-history scenarios.
>
> |                 | Metrics | History len = 1024 | History len = 2048 |
> | --------------- | ------- | ------------------ | ------------------ |
> | **BookReview**  | ROUGE-1 | 0.1978             | 0.2318             |
> |                 | ROUGE-L | 0.1071             | 0.1270             |
> |                 | METEOR  | 0.1859             | 0.2580             |
> |                 | BLEU    | 3.0821             | 3.5718             |
> | **MovieReview** | ROUGE-1 | 0.1888             | 0.2015             |
> |                 | ROUGE-L | 0.0977             | 0.1041             |
> |                 | METEOR  | 0.1598             | 0.1918             |
> |                 | BLEU    | 1.9795             | 2.3845             |
> | **CDReview**    | ROUGE-1 | 0.1870             | 0.1976             |
> |                 | ROUGE-L | 0.0938             | 0.0998             |
> |                 | METEOR  | 0.1540             | 0.1805             |
> |                 | BLEU    | 1.6705             | 1.9676             |
>
>
>
> **Q4: A discussion of potential confounders and how their presence might affect the proposed method would add important context to the paper’s causal claims.**
>
> **A4:** To address these concerns, we first follow Reviewer dbbY’s suggestion and conduct a stratification analysis, demonstrating that our method is robust to factors such as item popularity (potential confounders). If additional confounders exist (denoted by Z)—which is more likely on the data side, we just need to estimate the DCE as follows (according to [1]):
> $$\text{DCE} = \sum_{Z=z} [P(Y_t \mid U = u, x, z) - P(Y_t \mid U = 0, x, z)]p(z)$$
> This adjustment accounts for the main part of the method that could be affected by the confounder Z.
>
> [1] Pearl, Judea. Causality. Cambridge university press, 2009.

---

> ### Author Response · Authors · 2025-11-27
> **Looking Forward to Your Feedback**
>
> Dear Reviewer y5Zw,
>
> Thank you for your insightful review. We have added further experiments and analyses addressing all your points. If you have any additional suggestions or questions, please feel free to let us know.
>
> We appreciate your time and consideration.
>
> Best regards, The Authors

---

> ### Comment · Reviewer_y5Zw · 2025-11-28
> **Thanks for your reply**
>
> Dear Authors,
>
>  Thank you so much for addressing my concerns. I will increase my score since the proposed preference causal disentanglement is indeed interesting.
>
>  (1) For the information leakage, I have double-checked the literature, and indeed, the first round of training is only to provide the importance of how different tokens in the data can capture the user preference. Therefore, there is no information leakage.
>
>  (2) The additional experiments of different levels of user interactions help further demonstrate the generalizability of this method.
>
>  In addition, it would be of great interest if the visualization of the token importance could be attached in the final revision. For example, the DCE difference heatmap across different tokens and whether those differences can intuitively reflect the effect by context or the effect by user preference. Moreover, one more question:
>
>  (1) The current data-level importance depends on DCE, which further depends on a pretrained model. Therefore, if the model is not trained to capture the relation between preference and corresponding token (e.g., what if the language model never utilizes that history part, then after removing, the difference would be 0), leveraging the difference might not well reflect the ground-truth importance. One suggestion is to leverage the model ensemble difference there. Looking forward to more discussion on this regard.
>
> Thank you!

---

> ### Author Response · Authors · 2025-12-03
> **Response to Reviewer y5Zw**
>
> Dear Reviewer y5Zw,
>
> Thank you very much for your encouraging feedback. We are glad that the clarification on information leakage addresses your concerns.
>
> Regarding your helpful suggestion on visualization, we agree that illustrating token-level importance would provide more intuition. In the revision, in Appendix F and Figures 10, 11, 12,  we have included visualizations (e.g., heatmaps of DCE differences across tokens) for all three datasets to better show how the learned importance patterns reflect preference-driven behavior rather than contextual artifacts.
>
> In response to the additional question: our method is not applicable when there are no differences. In such cases, the most straightforward approach is to fine-tune the model on this dataset to compute the DCE or to use the in-training NextQuill model to compute the DCE directly. For a detailed discussion and the corresponding experimental results, please refer to our response to Reviewer hwTr (see W2&A2).
>
> Thank you again for the thoughtful comments and for increasing your score—we genuinely appreciate your support.

---

### Official Review · Reviewer_4wx3 · 2025-11-01

**Soundness:** 2
**Presentation:** 3
**Contribution:** 3
**Rating:** 4
**Confidence:** 4

**Summary:**

The paper introduces NextQuill, a LLM personalization framework that leverages causal graph to identify the more informative tokens in the user history. It estimates preference-driven components on the model side by intervening on user history and measuring resulting prediction changes, and on the data side by scoring tokens according to how strongly they reflect user-specific characteristics. NextQuill finetunes the inference LM on the weighted next token loss to align with these identified tokens. The experiment demonstrates mostly positive results for the proposed approach.

**Strengths:**

S1. Generally, I like the overall idea the authors present in the paper; in the user history, there are important information for personalization, and there is less relevant information. The idea is intuitive, and causal models can be naturally applied to the topic.

S2. The disentanglement between the model-side causal graph and user-side causal graph is also innovative and make good technical contribution.

S3. Although questions remain, the experiments include sufficient state-of-the-art baselines and commonly used datasets for a fair comparison.

**Weaknesses:**

W1. Scope of the paper.
LM Personalization can mainly be categorized into two categories: Personalized text generation [1,2] and personalized preference alignment [3], which are different in terms of the tasks. The paper claims to solve both as L124 claims “…representing text written or liked by the user.” However, the evaluation and experiments do not support the claim on the preference alignment as the author includes no preference optimization baselines [3]. Instead, it focuses on the text generation tasks such as review writings as seen in prior works on personalized text generation [1, 2]. We should not mix these concepts and it can create potential confusion to the readers. Please include more discussion in the related work and clearly call out the scope of the paper.

W2. The Identified Components.
The central claim of the paper is that causal graphs can help identify crucial language component for the task of personalization. However, in the experiments, we do not see the components that are identified to be idiosyncratic for user preference. Although Table 2 in the ablation studies demonstrate the effectiveness of including the causal loss in the model training, it does not exhibit the soundness of the causal graph as the improvement can simply come from better SFT alignment.

W3. LLM-as-a-Judge.
LLM-as-a-judge has recently been used in personalization evaluation, as seen in [4, 5]. In D.4, we appreciate the author for including the results for LLM-as-a-Judge. What is the scale for the evaluation? The delta seems incredibly large comparing to the lexical metrics such as BLEU and ROUGE. Some analysis and prompt examples would be appreciated. Additionally, LLM-TRSR seems to outperform the other models in LLM-as-a-Judge metric, but is one of the worst performers (0.0465) in traditional lexical metrics (ROUGE-1). I think we need more exploration and analysis into the results. Maybe even including more advanced peer-reviewed LLM-as-a-Judge that is designed for personalization [5].

**Questions:**

1. Please revisit the scope of the paper and include proper introduction for personalized preference optimization and personalized text generation. Consider refine the problem definition in L124 and carefully improve the definition.

2. Include the analysis on the proposed causal graph and justify its soundness through experiments. In addition to the ablation study results in Table 2, include the example of identified tokens and their corresponding influence in the SFT. Since we have two causal models, we should compare the effects of including the individual causal component in the experiment.

3. More analysis on the LLM-as-a-judge results. What is the experiment setting? Prompt of the LLM-as-a-judge?

4. Compare the performance of LLM-as-a-judge results and traditional lexical results. Especially for LLM-TRSR and the proposed models. What makes the LLM-as-a-Judge so different?

5. Experiment with LLM-as-a-judge for personalization? [5]

6. Please discuss and include more relevant works, e.g., [1, 2, 3, 6]

In general, I like the idea and would increase my score conditioned on the addressing of the above mentioned points.

[1] LaMP: When Large Language Models Meet Personalization. Alireza Salemi, et al.
[2] LongLaMP: A Benchmark for Personalized Long-form Text Generation. Kumar, et al.
[3] Personalized Language Modeling from Personalized Human Feedback. Xinyu Li, et al.
[4] Reasoning-Enhanced Self-Training for Long-Form Personalized Text Generation. Salemi, et al.
[5] ExPerT: Effective and Explainable Evaluation of Personalized Long-Form Text Geneartion. Salemi, et al.
[6] A Personalized Conversational Benchmark: Towards Simulating Personalized Conversations. Li, et al.

---

> ### Author Response · Authors · 2025-11-22
> **Response to Reviewer 4wx3**
>
> Dear Reviewer 4wx3,
>
> Thank you for the thoughtful comments. Below we provide our detailed responses, and we hope they address your concerns.
>
> **W1 & Q1: Clarifying Scope**
>
> **A1:** We thank the reviewer for raising this important distinction. We will revise the corresponding sections to clearly define our scope as personalized text generation, rather than personalized preference optimization. We will explicitly state in the revision:
>
> “We study personalized text generation, which aims to enhance an LLM by incorporating user-specific information to generate responses tailored to individual users. The ground-truth response 𝑦 denotes text written by the user.”
>
> This clarification reduces conceptual ambiguity and cleanly separates our setting from personalized preference alignment.
>
>
>
> **W2 & Q2: Ablation Study: Since we have two causal components, we should compare the effects of including each individually. The improvement could simply result from better SFT alignment.**
>
> **A2:** We have conducted ablation studies to isolate the contributions of each causal component.
> To better explain our ablations, we first clarify how our loss designs correspond to the defined causal effects,
>
> * **Eq. 6 loss:** SFT loss with highlighted preference-driven tokens identified by DCE (captures **DCE**).
> * **Eq. 7 loss:** MCE alignment loss that encourages the model’s internal causal preference effects to match labels, also highlighting preference-driven tokens (align **MCE and DCE**).
> * **Total loss:** **Eq. 6 + Eq. 7**.
>
> Based on this formulation, we provide the following ablations:
>
> - **MCE only:** SFT loss + Eq. 7 *without* highlighting preference-driven tokens ($\omega_t=1$ always; isolates **MCE**).
> - **DCE only:** Eq. 6 only (isolates **DCE**).
> - **MCE–DCE alignment:** SFT loss + Eq. 7 (captures **MCE-DCE alignment**).
>
> As shown in the table below, each ablation variant outperforms the base model (BaseSFT) and strong baseline (ContextSFT), confirming that each causal component contributes to the model’s performance.
>
>
> |                         | BookReview |         | MovieReview |         | CDReview |         |
> | ----------------------- | ---------- | ------- | ----------- | ------- | -------- | ------- |
> |                         | ROUGE-1    | ROUGE-L | ROUGE-1     | ROUGE-L | ROUGE-1  | ROUGE-L |
> | ContextSFT              | 0.1661     | 0.0836  | 0.1573      | 0.0753  | 0.1505   | 0.0714  |
> | BaseSFT                | 0.0752     | 0.0351  | 0.0620      | 0.0305  | 0.0668   | 0.0314  |
> | + MCE only | 0.1827     | 0.0907  | 0.1629      | 0.0796  | 0.1552   | 0.0744  |
> | + MCE-DCE alignment | 0.1876     | 0.0961  | 0.1671      | 0.0817  | 0.1672   | 0.0793  |
> | + DCE only     | 0.1958     | 0.1122  | 0.1865      | 0.0953  | 0.1805   | 0.0922  |
> | NextQuill               | 0.2318     | 0.1270  | 0.2015      | 0.1041  | 0.1976   | 0.0998  |
>
>
>
> **W3 & Q3 & Q4 & Q5 — Explain the LLM-as-a-Judge Evaluation:**
>
> **A3:** Thank you for the insightful comment. We were also initially puzzled by why LLM-as-evaluation results differed substantially from traditional evaluations. Following your suggestion, we adopted the recommended advanced LLM-as-Judge method ExPerT to re-run the evaluation. The updated results are summarized below. As shown, the outcomes are now better aligned with traditional methods — the differences are smaller, and LLM-TRSR is no longer consistently the best-performing across datasets. These results suggest that our previous evaluation prompt and settings may not have been suitable enough. Thanks for the suggestion again.
>
> | Methods    | BookReview | MovieReview | CDReview |
> | ---------- | ---------- | ----------- | -------- |
> | Qwen       | 1.0725     | 1.0260      | 1.0382   |
> | Contriever | 1.3513     | 1.2682      | 1.2972   |
> | LatestK    | 1.3544     | 1.2284      | 1.2834   |
> | LLM-TRSR   | 1.5236     | 1.4545      | 1.6977   |
> | CoS        | 1.1987     | 1.1032      | 1.1334   |
> | SFT        | 1.1041     | 1.0452      | 1.0895   |
> | ContextSFT | 1.7911     | 1.4664      | 1.6157   |
> | NextQuill  | 2.1487     | 1.6984      | 1.9267   |
>
> **Q6: Please discuss and include more relevant works, e.g., [1, 2, 3, 6]**
>
> **A6:** Thank you for the suggestion. We will include these references and discuss them in the revised manuscript and upload the revised paper later.

---

> ### Author Response · Authors · 2025-11-27
> **Looking Forward to Your Feedback**
>
> Dear Reviewer 4wx3,
>
> Thank you for your insightful review. We have added further experiments and analyses addressing all your points. If you have any additional suggestions or questions, please feel free to let us know.
>
> We appreciate your time and consideration.
>
> Best regards, The Authors

---

### Official Review · Reviewer_hwTr · 2025-11-06

**Soundness:** 3
**Presentation:** 2
**Contribution:** 3
**Rating:** 6
**Confidence:** 4

**Summary:**

This work presents a framework for personalization, positing that alignment should target causal preference effects rather than uniformly fitting all tokens in a sequence. The method defines two key concepts: 1) a Model-side Causal Effect (MCE), which quantifies the change in next-token predictions when user history ($H$) is included versus omitted; and 2) a Data-side Causal Effect (DCE), which uses a separate model ($f_{\theta_D}$) exposed to the dataset $D$ to estimate which ground-truth tokens were likely driven by user preferences. The training objective combines two components: a preference-weighted cross-entropy loss (which up-weights tokens identified as DCE-positive) and a causal preference alignment loss. This second loss explicitly encourages the model's MCE to align with the ground-truth tokens identified by the DCE. Empirically, the proposed method reports consistent performance gains over retrieval-based and PEFT baselines across two datasets. This improvement is achieved with additional training-time overhead but adds no extra cost at inference.

**Strengths:**

The paper provides a clear motivation and an intuitive conceptual framing. It proposes a causal framework that moves beyond the naive approach of uniformly aligning all tokens, instead targeting preference-driven components. This causal perspective for personalization and its instantiation are valuable contributions.

The empirical evaluation is thorough, reporting consistent and significant performance gains over strong baselines across several datasets. The results also compellingly show that these improvements scale favorably with increasing model size.

The ablation studies effectively demonstrate the utility of core components (the preference-weighted token loss and the causal alignment loss), indicating that they provide complementary contributions to the final performance.

**Weaknesses:**

This method can fail when the main preference signal lives in the current query/context $x$. Because it defines “preference-driven” via the with-history vs. no-history contrast and up-weights only $H$-mediated tokens, it systematically down-weights $x$-mediated cues (e.g., “be brief,” tone, formatting). So in cases where $x$ actually carries strong preference evidence, the method can misprioritize or even suppress those signals—i.e., my concern is that it doesn’t work properly when on-the-fly preferences are expressed in $x$.

DCE hinges on f_{\theta_D}—a model “that has seen D”—to identify which tokens are “preference-driven.” However, the assumption that merely having seen D suffices to use f_{\theta_D} as a reliable approximator is not fully justified. Given that the manuscript does not specify what f_{\theta_D} actually is (backbone, training objective, whether it is causally trained, and its personalization performance), I have reservations about the fidelity of Eq. (5). If f_{\theta_D} is causally trained, the paper should describe the objective, explain how it differs from NextQuill, and include f_{\theta_D} as a baseline. If f_{\theta_D} is not causally trained yet already strong at personalization, that weakens the claim that causal effects are necessary and calls for clarification of NextQuill’s added value beyond simply using f_{\theta_D} for personalization. If f_{\theta_D} is not strong, using it to approximate DCE is unreliable. To resolve these concerns, the manuscript should precisely specify f_{\theta_D}, report its personalization metrics, and provide stronger validation that f_{\theta_D} can serve as a reliable approximator for DCE (e.g., small human checks). As written, the support is currently insufficient.

**Questions:**

Please see weaknesses.

Typo: “personalizaiton” (L134)

---

> ### Author Response · Authors · 2025-11-22
> **Response to Reviewer hwTr**
>
> Dear Reviewer hwTr,
>
> We sincerely appreciate your valuable comments. Below we provide our detailed responses, and we hope they address your concerns.
>
> **W1: When the query/context $x$ carries strong preference evidence, the method can misprioritize or even suppress those signals**
>
> **A1:** Thank you for the question. We provide two points in response. First, in our design, the method still includes the next-token prediction loss, which means the prompt information is utilized and can influence the model’s generation results. Therefore, the preference signals in prompts are expected to be leveraged. Additionally, if we want to further emphasize the effect of preference information in the prompt, it can be treated as part of the historical input. In this way, the prompt information can play a role similar to history in our preference modeling framework, requiring only a slight extension of our method.
>
> **W2:  Specify $f_{\theta_D}$ can serve as a reliable approximator for DCE**
>
> **A2:** Thank you for raising this valuable point. In fact, we have considered different implementations:
>
> 1. The first approach dynamically computes weights (DCE) based on NextQuill itself during training, aiming to progressively identify preference-driven tokens more accurately. This is our default method.
> 2. We also experimented with using a small, pre-trained NextQuill model (1.5B, trained via dynamic-weight training) to compute fixed weights (a weaker $f_{\theta_D}$), which were subsequently used to train a 3B NextQuill model. We then compared the personalization performance of the 1.5B NextQuill with that of the resulting 3B NextQuill, as summarized in the table below. The 1.5B NextQuill achieves comparable performance to the strongest baseline 3B ContextSFT, but it performs significantly worse than the 3B NextQuill. These results indicate that using a small pre-trained NextQuill to compute weights is a viable approach: the weight-computing model is not too weak, while applying the larger NextQuill on top yields substantial improvements over the weight-computing model.
>
> We consider both approaches reasonable; however, the second method involves additional training cost, so we use the first method as the default.
>
>
>
> | Dataset         | Metrics | ContextSFT   3B | NextQuill 1.5B (dynamic DCE) | NextQuill 3B (dynamic DCE) | NextQuill 3B (static DCE from NextQuill 1.5B) |
> | --------------- | ------- | --------------- | ---------------------------- | -------------------------- | --------------------------------------------- |
> | **BookReview**  | ROUGE-1 | 0.1661          | 0.1631                       | 0.2318                     | 0.2549                                        |
> |                 | ROUGE-L | 0.0836          | 0.0938                       | 0.1270                     | 0.1432                                        |
> |                 | METEOR  | 0.2158          | 0.2282                       | 0.2580                     | 0.2892                                        |
> |                 | BLEU    | 2.1214          | 2.7469                       | 3.5718                     | 4.3349                                        |
> | **MovieReview** | ROUGE-1 | 0.1573          | 0.1251                       | 0.2015                     | 0.2107                                        |
> |                 | ROUGE-L | 0.0753          | 0.0637                       | 0.1041                     | 0.1070                                        |
> |                 | METEOR  | 0.1718          | 0.1512                       | 0.1918                     | 0.2085                                        |
> |                 | BLEU    | 1.7151          | 1.6808                       | 2.3845                     | 2.6761                                        |
> | **CDReview**    | ROUGE-1 | 0.1505          | 0.1326                       | 0.1976                     | 0.2052                                        |
> |                 | ROUGE-L | 0.0714          | 0.0653                       | 0.0998                     | 0.1019                                        |
> |                 | METEOR  | 0.1583          | 0.1547                       | 0.1805                     | 0.1954                                        |
> |                 | BLEU    | 1.3487          | 1.4181                       | 1.9676                     | 2.2284                                        |

---

> ### Author Response · Authors · 2025-11-27
> **Looking Forward to Your Feedback**
>
> Dear Reviewer hwTr,
>
> Thank you for your insightful review. We have added further experiments and analyses addressing all your points. If you have any additional suggestions or questions, please feel free to let us know.
>
> We appreciate your time and consideration.
>
> Best regards, The Authors

---

### Official Review · Reviewer_dbbY · 2025-11-10

**Soundness:** 3
**Presentation:** 2
**Contribution:** 3
**Rating:** 6
**Confidence:** 3

**Summary:**

This paper introduces NextQuill, a causal preference–modeling framework for LLM personalization. The key idea is to focus training on preference-driven tokens in both the targets and the model’s predictions. Concretely, the authors compute model-side and data-side causal preference effects and fine-tune the model with (i) a preference-weighted objective and (ii) an effect-alignment loss. Experiments on four personalization tasks show consistent improvements over retrieval-based and PEFT baselines, supported by ablations.

**Strengths:**

- **Clear, principled formulation.** Personalization is framed as token-level causal effects, which naturally motivates a two-loss fine-tuning scheme (preference-weighted next-token loss + effect-alignment loss). The causal lens is intuitive and likely useful beyond the evaluated tasks.

- **Effective performance with solid analysis.** The method achieves strong gains over retrieval and PEFT baselines across Amazon review domains and a long-form topic-writing task. Ablations indicate both components (token weighting and effect alignment) contribute meaningfully.

**Weaknesses:**

- **Limited validation of the causal preference effects.**
Qualitative illustrations are helpful, but deeper evidence is needed that tokens flagged as “preference-driven” correspond to user preferences rather than context artifacts.
Suggestion: Add systematic analyses—e.g., human-annotated token studies; correlation between model-side and data-side effects; negative-control tests—to show the method truly captures preference vs context.

- **Gap between formulation and real-world.**
User histories often entangle preference with contextual factors (topic, recency, item attributes).
Suggestion: Report robustness under noisy or partially mismatched histories; perturb or mask contextual attributes to quantify spillover from context to preference labels.

- **Intervention vs conditioning (identifiability).**
The method effectively treats do(𝐻=ℎ) as equivalent to conditioning on H=h. In real data, unobserved confounders (item popularity, seasonality) can bias effect estimates.
Suggestion: Provide a clearer justification and a sensitivity/backdoor analysis (e.g., synthetic confounders, stratification by proxies) to support the causal claims.

**Questions:**

**Multiple or conflicting preferences.** Can the framework handle composite or conflicting user preferences (e.g., time-varying or multi-persona) and decompose effects by type?

**Context in histories.** How do you separate preference from context when histories include both? Any diagnostics showing that preference-driven tokens are not merely contextual artifacts?

I

---

> ### Author Response · Authors · 2025-11-22
> **Response to Reviewer dbbY (1/2)**
>
> Dear Reviewer dbbY,
>
> Thank you for your valuable comments. Below we provide our detailed responses, and we hope they address your concerns.
>
> **W1: Add systematic analyses (e.g., negative-control tests) to validate the causal preference effects (tokens flagged as “preference-driven”)**
>
> **A1:** Following the suggestion, we conducted a negative-control test by randomly selecting tokens for re-weighting while keeping the weighting mechanism unchanged. As summarized in the table below, replacing the detected preference-driven tokens with random tokens leads to a substantial performance drop, confirming the effectiveness of the causal preference effects.
>
> |             |                   | ROUGE-1 | ROUGE-L | METEOR | BLEU   | BERTScore |
> | ----------- | ----------------- | ------- | ------- | ------ | ------ | --------- |
> | BookReview  | NextQuill_NegCtrl | 0.1811  | 0.0909  | 0.2170 | 2.2684 | 0.8024    |
> |             | NextQuill         | 0.2308  | 0.1274  | 0.2573 | 3.3798 | 0.8182    |
> | MovieReview | NextQuill_NegCtrl | 0.1569  | 0.0770  | 0.1628 | 1.8237 | 0.7947    |
> |             | NextQuill         | 0.2009  | 0.1032  | 0.1909 | 2.2701 | 0.8064    |
> | CDReview    | NextQuill_NegCtrl | 0.1516  | 0.0730  | 0.1492 | 1.4167 | 0.7878    |
> |             | NextQuill         | 0.1916  | 0.0973  | 0.1740 | 1.7313 | 0.8060    |
>
> **W2 and Q2: User histories often entangle preference with contextual factors (topic, recency, item attributes).**
>
> **A2:** In general, user preferences span multiple aspects, including semantics, writing style, and emotional tone [1]. Since semantics is inherently coupled with contextual factors (e.g., topics and item attributes), fully disentangling them would remove part of the preference information, which is inconsistent with our goal.
>
> To address the concern that performance may stem from context rather than more accurate preference extraction: 1) Baselines also leverage the same contextual information, ensuring a fair comparison; 2)  In addition, our model captures preference dimensions beyond context. For example, some users consistently include phrases like “All opinions are my own” in their reviews — these stylistic patterns can be identified by our method as preference-driven, not context-driven.
>
> [1] Qiu et.al., Measuring what makes you unique: Difference-aware user modeling for enhancing llm personalization. ACL 2025.

---

> ### Author Response · Authors · 2025-11-22
> **Response to Reviewer dbbY (2/2)**
>
> **W3: Potential confounders (e.g., item popularity, seasonality)**
>
> **A3:** To address this concern, we conducted a stratification analysis based on item popularity, approximated by item frequency. Items were divided into two groups (Low Popularity & High Popularity), and modeling was performed separately for each. As shown in the table below, our method consistently obtains performance gains across both groups, suggesting our method is robust to item popularity. Regarding seasonality, the data do not allow for reliable estimation, making experimental analysis difficult. However, since our tasks focus on review generation and topic writing, user behavior is expected to remain largely stable across seasons, as these preferences are relatively long-term and primarily driven by content rather than temporal factors.
>
> | Datasets        | Metrics   | ContextSFT (Low) | NextQuill (Low) | ContextSFT (High) | NextQuill (High) |
> | --------------- | --------- | ---------------- | --------------- | ----------------- | ---------------- |
> | **BookReview**  | ROUGE-1   | 0.1651           | 0.2079          | 0.1834            | 0.2336           |
> |                 | ROUGE-L   | 0.0826           | 0.1138          | 0.0919            | 0.1292           |
> |                 | METEOR    | 0.2149           | 0.2442          | 0.2406            | 0.2632           |
> |                 | BLEU      | 2.0779           | 3.0924          | 2.3964            | 3.5014           |
> |                 | BERTScore | 0.8011           | 0.8114          | 0.8075            | 0.8187           |
> | **MovieReview** | ROUGE-1   | 0.1588           | 0.1785          | 0.1748            | 0.1946           |
> |                 | ROUGE-L   | 0.0750           | 0.0939          | 0.0837            | 0.1009           |
> |                 | METEOR    | 0.1745           | 0.1724          | 0.1955            | 0.1845           |
> |                 | BLEU      | 1.6156           | 1.9791          | 1.9847            | 2.0724           |
> |                 | BERTScore | 0.7956           | 0.8006          | 0.8036            | 0.8087           |
>
>
>
> **Q1: Can the framework handle composite or conflicting user preferences (e.g., time-varying or multi-persona)?**
>
> **A1:** This is an interesting question. For our method, the extracted preferences are determined by the historical input provided to the LLM. The model does not distinguish between different preferences within the history. To handle conflicts, additional control over the input history is required. For example, in the time-varying scenarios, capturing a user’s *current* preferences calls for preserving recency in the input sequence, whereas reflecting *stable* preferences that persist across the entire history requires incorporating longer historical context.

---

> ### Author Response · Authors · 2025-11-27
> **Looking Forward to Your Feedback**
>
> Dear Reviewer dbbY,
>
> Thank you for your insightful review. We have added further experiments and analyses addressing all your points. If you have any additional suggestions or questions, please feel free to let us know.
>
> We appreciate your time and consideration.
>
> Best regards, The Authors

---

### Author Response · Authors · 2025-12-03
**To AC: Summary of Reviews and Discussion**

Dear Area Chair,

We sincerely appreciate your meta-review of our paper, especially given the substantial additional workload caused by the OR leaking. For your convenience, we provide a concise summary of the reviews and the discussion.

To the best of our knowledge, this work is **the first** to effectively integrate causal reasoning with large language models for personalization, achieving substantial improvements across multiple tasks and datasets. We are grateful that the reviewers recognized this contribution.

Our paper initially received scores of 6, 6, 4, and 4.
- For one of the two 4-score reviewers (y5Zw), the primary concern resulted from **a misunderstanding related to information leakage**; **after clarification, the reviewer agreed to raise the score**. The remaining requirements for this reviewer have also been fully satisfied.
- The other 4-score reviewer (4wx3) indicated in the initial review that our paper is interesting and **that the score would be raised once specific concerns were resolved**. This reviewer focused on three aspects: (a) the results of LLM-as-a-Judge evaluation; (b) the need for additional ablations; and (c) clearer scope boundaries for personalized text generation. We addressed (a) and (b) with new experiments and revised the paper to clarify (c) as requested. Unfortunately, **we did not receive feedback due to the early system closure** resulting from the OpenReview leak, which prevented the reviewer from responding; however, **all concerns have been addressed**.
- We also provided comprehensive responses to both positive reviews.

Below is a consolidated summary of the reviewers’ major concerns and our responses.

**Concerns:**

(1) validity and robustness of the estimated causal preference effects,

(2) potential entanglement between user preferences and contextual factors in user history,

(3) influence of possible confounders (e.g., popularity) on causal estimation,

(4) scope clarification for personalized text generation versus preference optimization, and

(5) evaluation questions, including ablations and the LLM-as-a-Judge methodology.

**Our responses:**

(1) Causal validity and robustness.
Following the reviewers’ suggestions, we conducted a negative-control test by reweighting random tokens instead of preference-driven ones. This resulted in clear performance degradation across all datasets, confirming that the detected causal preference effects are meaningful. We also added token-level heatmaps illustrating interpretable preference patterns.

(2) Preference–context disentanglement.
We clarified that user preferences naturally span semantics, style, and tone, making the complete disentanglement between context (semantics) and preferences neither meaningful nor achievable. Additional case studies demonstrate that our method captures preference-driven stylistic cues beyond contextual content.

(3) Potential confounders.
We performed a stratification analysis based on item popularity, showing that our method consistently improves performance across all popularity levels. That means our method is robust to popularity. We further discussed possible adjustments, following standard causal-inference formulations, that can be applied if confounders are indeed present.

(4) Scope clarification.
We revised the Problem Formulation section to explicitly define our scope as personalized text generation, avoiding ambiguity with personalized preference optimization.

(5) Evaluation methodology.
We expanded our ablations to isolate the contributions of both MCE and DCE, verifying that each component individually enhances personalization. For evaluation, we reran LLM-as-a-Judge experiments using the stronger ExPerT framework, which produced results more closely aligned with traditional metrics, solving the inconsistency issue.

Overall, we believe the new analyses and clarifications directly address all reviewer concerns. We sincerely hope the AC will consider these in the final evaluation. Our work offers a novel causal framework for modeling token-level preference effects and contributes meaningful insights to the emerging direction of LLM personalization.

*We have incorporated all rebuttal content and additional experiments into our revised paper, marking all changes in blue*.

Best regards,

The Authors

---

### Meta-Review · Area_Chair_zXgv · 2026-01-05

**Summary:**

This paper proposes a new approach to LLM personalization. The key idea is to estimate the effect of user histories and preferences on next-token prediction and use them as weights in supervised fine-tuning (SFT). The approach is motivated by causal modeling and simplified so that it can be implemented in LLMs. It is evaluated empirically on 3 recommender system problems and compared to 9 baselines. This is a solid paper, with both theory and good experiments, which will motivate more future work. The reviewers had many questions and suggestions, some of which I list below:

* **Are the personalized tokens causal?** The authors conducted a random permutation test to show it.

* **Does popularity play a role?** The authors showed that personalization improves recommendations of both popular and less popular items.

* **Focus on text personalization only:** The authors clarified it.

* **LLM-as-a-judge not aligned with other metrics:** The authors improved the evaluation using personalized LLM judges.

This was an exemplary response, where one reviewer increased their score and another one would likely do it. Therefore, I have full confidence in recommending acceptance of this paper.

My additional comment is that the math in Section 3 is only loosely connected. Please make this clear. The personalization effects that you estimate are similar to [ATE and CATE](https://en.wikipedia.org/wiki/Average_treatment_effect). This may be a better motivation and starting point for your work than using the do calculus.

**Reviewer Concerns:**

This was an exemplary response, where one reviewer increased their score and another one would likely do it. All concerns were addressed briefly and to the point, with additional experiments if needed.

**Reviewer Scores:**

Reviewer y5Zw increased their score, from 4 to at least 6. Reviewer 4wx3 promised to increase their score, from 4 to at least 6, conditioned on the rebuttal. The rebuttal to Reviewer 4wx3 addressed their concerns. With these score changes, the paper would be in the acceptance territory.

---

### Decision · Program_Chairs · 2026-01-26

Accept (Poster)